# Current Trends on Seaweeds: Looking at Chemical Composition, Phytopharmacology, and Cosmetic Applications

**DOI:** 10.3390/molecules24224182

**Published:** 2019-11-18

**Authors:** Bahare Salehi, Javad Sharifi-Rad, Ana M. L. Seca, Diana C. G. A. Pinto, Izabela Michalak, Antonio Trincone, Abhay Prakash Mishra, Manisha Nigam, Wissam Zam, Natália Martins

**Affiliations:** 1Student Research Committee, Bam University of Medical Sciences, Bam 4340847, Iran; bahar.salehi007@gmail.com; 2Zabol Medicinal Plants Research Center, Zabol University of Medical Sciences, Zabol 61615-585, Iran; 3cE3c- Centre for Ecology, Evolution and Environmental Changes/Azorean Biodiversity Group & University of Azores, Rua Mãe de Deus, 9501-801 Ponta Delgada, Portugal; ana.ml.seca@uac.pt; 4QOPNA & LAQV-REQUIMTE, Department of Chemistry, University of Aveiro, 3810-193 Aveiro, Portugal; diana@ua.pt; 5Department of Advanced Material Technologies, Faculty of Chemistry, Wroclaw University of Science and Technology, Smoluchowskiego 25, 50-372 Wroclaw, Poland; izabela.michalak@pwr.edu.pl; 6Institute of Biomolecular Chemistry, Consiglio Nazionale delle Ricerche, 80078 Pozzuoli, Naples, Italy; antonio.trincone@icb.cnr.it; 7Department of Pharmaceutical Chemistry, Hemvati Nandan Bahuguna Garhwal University, Srinagar Garhwal-246174, Uttarakhand, India; abhaypharmachemhnbgu@gmail.com; 8Department of Biochemistry, Hemvati Nandan Bahuguna Garhwal University, Srinagar Garhwal-246174, Uttarakhand, India; anandmanisha23@gmail.com; 9Department of Analytical and Food Chemistry, Faculty of Pharmacy, Al-Andalus University for Medical Sciences, Tartous, Syria; 10Department of Medicine, Faculty of Medicine, University of Porto, Alameda Prof. Hernâni Monteiro, 4200-319 Porto, Portugal; 11Institute for Research and Innovation in Health (i3S), University of Porto, 4200-135 Porto, Portugal

**Keywords:** seaweeds, primary metabolites, secondary metabolites, bioactivity, pharmaceutical and cosmetic applications

## Abstract

Seaweeds have received huge interest in recent years given their promising potentialities. Their antioxidant, anti-inflammatory, antitumor, hypolipemic, and anticoagulant effects are among the most renowned and studied bioactivities so far, and these effects have been increasingly associated with their content and richness in both primary and secondary metabolites. Although primary metabolites have a pivotal importance such as their content in polysaccharides (fucoidans, agars, carragenans, ulvans, alginates, and laminarin), recent data have shown that the content in some secondary metabolites largely determines the effective bioactive potential of seaweeds. Among these secondary metabolites, phenolic compounds feature prominently. The present review provides the most remarkable insights into seaweed research, specifically addressing its chemical composition, phytopharmacology, and cosmetic applications.

## 1. Introduction

Seaweed is a colloquial term for macroscopic, multicellular benthic marine algae. Seaweeds are one of the largest producers of biomass in the marine environment [1], and constitute an important part of the diet and traditional medicine in many parts of Asia since prehistoric times [2].

Seaweeds belong to different kingdoms within the classification of living organisms and represent a potential reservoir of bioactive natural products. They grow in a wide range of sizes including more than 10,000 species of the fastest growing plants all over the world in a variety of shapes and colors [3]. Seaweeds are attached or freely floating primitive plants lacking true root, stem, and leaves, and constitute important marine living renewable resources. They grow in deep-sea areas up to 180 m in depth, in estuaries, and also in black water on solid substrates like pebbles, rocks, dead corals, shells, and plant material, and are attached to the bottom in comparatively shallow rocky coastal areas [4], especially where they are exposed at low tide [3], constituting one of the important living resources of the ocean.

Marine algae are classically divided into four groups, the prokaryotic Cianophyceae (blue green algae) and the eukaryotic Ochrophyta (brown algae), Rhodophyta (red algae), and Chlorophyta (green algae), which differ mainly in their pigmentary composition, nature of their cell walls, and reserve polysaccharides [5]. Being seaweeds, one of the primary producers and the base of the food chain in oceans [6,7], they play a crucial role in supporting marine biodiversity [8]. Seaweeds are also useful as indicators of environmental stresses [9] as the distribution and abundance of seaweeds is affected by exposure to disturbances like desiccation stress, high temperatures, and competition with coastal fauna and flora [6]. Alterations in global climate and its variability influences biological, ecological, and socioeconomic systems, therefore they also have a remarkable impact on the marine environment [10]. Although seaweeds are usually well adapted to their thermal environment, temperature influences the enzymatic and metabolic functioning of seaweeds [11] and could result in cellular and sub cellular damage [12], leading to slow growth and development [13].

Seaweeds provide the raw material for many industrial productions like seaweed gums [14], which are widely consumed as food in Asian countries, especially by coastal people in the form of edible seaweeds known as “sea vegetables” since the beginning of time [15], thus serving as an alternative source of dietary fiber, protein, and minerals. Moreover, they possess many secondary metabolites responsible for their bioactivities, thus having commercial application in the medical, pharmaceutical, nutraceutical, agricultural and cosmetic industries. Interestingly, in the past three decades, the discovery of potential metabolites and biological activities from seaweeds has increased remarkably, thanks to the advancement of modern research tools such as the cataloguing of marine chemicals, x-ray, and NMR activity. However, there are still numerous unexplored natural bioactive compounds that could serve as a source of novel bioactive compounds for pharmaceutical companies. In this sense, this review aims to provide updated insights into seaweed research, specifically addressing their chemical composition, phytopharmacology, and cosmetic applications.

## 2. Seaweeds: A Critical Perspective

### 2.1. Food and Nutritional Applications

Seaweeds have been cultivated and utilized directly as food for humans or as feed to produce food indirectly for human consumption since ancient times. As a staple diet item, it has been used in Japan, China, and Korea for ages [16] and have been reported to possess many essential nutrients such as vitamins, amino acids, minerals, omega-3 fatty acids, and several biologically active compounds [4]. Moreover, they are being recognized as a sustainable food source with the potential to play a major role in providing food security worldwide [17]. Although seaweeds are part of the diet in many Asian countries and some European nations, there are some challenges in terms of their incorporation into the general diet in many places globally. Seaweeds are mainly used as food in regions of coastal areas [18]. In Japan, the chief food species are Kombu (*Saccharina japonica*), Nori (*Pyropia spp*.), and Wakame (*Undaria pinnatifida*). Although in Western cuisine seaweeds are not used much, a renewed interest in Western countries is growing for its use as sea vegetables.

Seaweeds harbor numerous polysaccharides that are thrust out by the hydrocolloid industry (e.g., alginate and agar from brown and red seaweeds, respectively) [19]. Agar-agar, a gum discovered in Japan that is extracted from the *Gelidium* species (red seaweed), is a brilliant gel-forming substance due to the unusual length of its carbohydrate molecules. It has the ability to withstand near boiling-point temperatures, making it ideal for use in jellied preparations since the ingredients can be treated at high temperatures and then cooled [20]. Carrageenans are a family of linear sulfated polysaccharides widely used in the food industry and are extracted from red edible seaweeds. Seaweed carrageenan and agar are used in the food industry as thickening and gelling agents, and are also used as growth medium for microorganisms [21]. Moreover, minor polysaccharides found in the cell wall are fucoidans (from brown seaweeds), ulvans in green seaweeds, and xylans (from certain red and green seaweeds). Seaweeds also possess storage polysaccharides like laminarin in brown seaweeds and floridean starch in red seaweeds. Most of these polysaccharides are not digested by human intestinal bacteria and therefore can be regarded as dietary fibers [22]. Water-soluble and -insoluble fibers have been associated with different physiological effects. Many viscous soluble polysaccharides have been reported to possess hypocholesterolemic and hypoglycemic effects [23]. Fucoidans have been particularly studied among these polysaccharides and found to possess remarkable bioactive functions (anti-proliferative, anticancer, anti-coagulant, anti-thrombotic, anti-viral, anti-inflammatory, and anti-complementary agent) [2]. Seaweeds obtain a stupendous treasure of minerals, macro, and trace elements from the sea. Interestingly, the mineral fraction of some seaweeds accounts for up to 36% of dry matter that varies with species, season, wave exposure, environmental and physiological influences, and the processing and mineralization methods [24,25]. Additionally, seaweeds are remarkable sources of vitamins A, B, C, and E, and minerals. It has been reported that the vitamin C content of Nori is 1.5 times higher than in oranges [18]. Furthermore, vitamin B is found in an ample amount in all seaweeds that belong to phaeophyceae, and brown seaweeds have traditionally been used for treating thyroid goiters [26].

The protein content of brown seaweeds is usually small, and higher protein contents have been documented in green and red seaweeds. Lipids, although representing only 1–5% of dry matter, show an interesting polyunsaturated fatty acid composition regarding omega 3 and omega 6 acids, which have implications in the prevention of cardiovascular diseases, osteoarthritis, and diabetes. It is imperative to note that green algae show interesting levels of alpha linolenic acid and that red and brown algae are especially rich in 20 carbon fatty acids (i.e., eicosapentaenoic acid and arachidonic acid).

### 2.2. Seaweeds as Biofertilizers

Seaweeds are available on the market as biofertilizers in different forms such as seaweed liquid fertilizers (SLF) and a powder form of seaweed manure [27]. This natural seaweed fertilizer may partially substitute the prevailing synthetic fertilizer. The various elements present in seaweed such as amino acids, macro and micro-nutrients, vitamins, auxins, cytokinin and abscisic acid (ABA) like growth substances can contribute to enhanced growth and crop yield via modulating metabolic pathways. Moreover, other beneficial effects of seaweed extract applications on plants such as improving water holding capacity and enhancement of the growth of beneficial soil microbes have also been documented. Eventually, all of these contribute to soil conditioning, leading to the promotion of root development, better nutrient uptake by the roots, early flowering and increased fruit formation and yield, and enhancing abiotic stress tolerance and defense against pests, diseases, and microorganisms [27].

## 3. Macroalgae Metabolites Diversity

Due to the diversity of constituents in seaweeds, they are a good supply of key compounds including carbohydrates, protein, lipids, and minerals as well as a rich source of health-promoting secondary metabolites capable of acting on a wide spectrum of disorders and/or diseases, and are excellent leading compounds in the development of new drugs and new additives in the food and cosmetic industries.

### 3.1. Primary Metabolites

Like other vegetal species, seaweeds produce several compounds by primary metabolism, called primary metabolites, which are essential to their growth, survival, and proliferation. These primary metabolites can be classified in several classes: lipids, proteins, and carbohydrates. In addition to the primary metabolites, algae accumulate minerals that are also essential to seaweed life and contribute to both its nutritional and pharmacological value. The seaweed’s primary metabolite content for each of these classes is quite variable and depends largely on the species, geographical area, and season [28,29,30]. The average content of the main primary metabolites’ classes in green, brown, and red seaweeds is shown in Table 1.

#### 3.1.1. Carbohydrates

Macroalgae’ carbohydrates are one of the primary metabolite classes whose chemical composition and structural characteristics are more variable and, at same time, what most differentiates the three groups of seaweeds. Although seaweed may contain free simple carbohydrates such as *Fucus spiralis* that have high amounts of the alcoholic sugar mannitol [48], they are chiefly polysaccharides, mostly non-starch and non-homopolymers that can be branched or linear, and involve monosaccharides (mainly aldohexoses and aldopentoses) where one or more hydroxyl group can be replaced by a sulfate or acetyl group, and the terminal CH_2_OH group can be replaced by methyl or carboxylic groups; monosaccharides that are bonded by a glycosidic linkage also have different possibilities (α or β, 1→3, 1→4, 1→6, …). As a result of the miscellaneous combination of these possibilities, seaweed polysaccharides are structurally highly differentiated and characteristic of each group of algae (Table 2). These polysaccharides have a high commercial value and important applications in the pharmaceutical and food industries [34,49,50]. The most well-known polysaccharides with industrial applications are the sulfated (fucoindans, carragenans, and ulvan) and non-sulfated (e.g., alginates and agars) polysaccharides. Some are less abundant, currently without industrial applications, and are still under investigation such as laminarin, xylans, porphyrans, argassan, and floridean [51].

The carbohydrates present in seaweeds correspond, on average, in all algae groups, to about 50% of the mass of dried algae (Table 1). However, they are mainly polysaccharides not degraded by mammalian enzymes, which mean that dietary fibers such as fucoidans, alginates, agar, and carrageenans (Table 2) also have a low caloric value [49]. These dietary fibers play a very important role in human health, contributing to regulating the intestinal tract, cholesterol, and blood sugar, and significantly to the prevention and treatment of various diseases such as cancer, obesity, and diabetes [33,52]. In addition, several studies have shown other pharmacological effects of these pure dietary fibers such as anti-inflammatory, anticoagulant, antibacterial, hypolipidemic, and prebiotic activities, among others [51]. The total dietary fiber content can range from 1.0 to 69% of the dried algae, whereas in green and red algae, about 50% of this value are soluble fibers like agar, alginates, or carrageenans, and reach an even higher percentage in brown algae [53].

##### Fucoidans

Fucoidans are a family of L-fucose polysaccharides with a variable percentage of sulfate groups and different glycosidic linkage, and are found in brown seaweeds and extracted mainly from *Ecklonia cava, Ascophyllum nodosum*, and *Undaria pinnatifida* species, whose content can range from 1.5 to 20% (dw), although the highest content has been found in *Fucus vesiculosus* [49,51]. Fucoidans, among then alginates and laminarans, are the most abundant polysaccharide in brown seaweeds. The content of fucoidans depends on the species, the geographical area, the season in which it is harvested, and environmental factors such as salinity and nutrients [54].

Like other seaweed polysaccharides, fucoidans have a structural function, similar to that exhibited by hemicelluloses in terrestrial plants. However, as seaweeds have very low content in lignin and cellulose [49], the seaweed structure is more flexible, which helps them stand strong against ocean waves and currents [51].

Several studies have reported the pharmacological interest of fucoidans since they exhibit anti-inflammatory, anticoagulant, antimicrobial, and antitumor activities, besides having an immuno-modulatory effect and could be used to treat liver and renal diseases [2,55,56,57,58,59,60]. However, these beneficial effects are dependent on the molecular weight, position, and content of the sulfate groups and the presence of other minor monosaccharides in the fucoidan structure [61,62]. Thus, the experimental conditions used in the isolation of fucoidans (more or less aggressive) are extremely relevant in obtaining polysaccharide rich extracts with their pharmacological properties intact.

Additionally, the use of fucoidans in cosmetic products has already been described in the literature, as they appear to attenuate the visible effects of skin ageing such as blemishes, wrinkles, and freckles [63,64,65].

##### Agars

Agars, along with carrageenans, are the most abundant structural polysaccharide in red seaweeds, and are identified mainly in the *Gracilaria* and *Gelidium* species in a content that can range from 21 to 42% (dw) [49,51]. Agars, also known as agar-agar, are water-soluble dietary fibers well known in Japanese cuisine as “kanten”, with extensive applications in food (90% of the agar produced globally) mainly as a gelling, thickening, and stabilizing food additive (E406), and is also used by the medical pharmaceutical industries (e.g., in the production of capsules and as a medium for cell and bacterial cultures) [49,66]. Some studies suggest that *Gracilaria gracilis* is the red algae with the highest agar content and with the best quality (gel strength, reversibility, color, flavor, pH stability, etc.) [66]. Other researchers have suggested that *Gelidium*-extracted agar typically has a better quality such as a higher gel strength [67]. However, several studies have shown that factors such as season and geographical area of harvest, algae storage and agar extraction methods, and eco-physiological factors cause variations in both the extraction yield and agar quality [28,68,69].

Agar is a general name for agarose and agaropectin, with agarose the most abundant (up to 70% of agar polysaccharide), while agaropectin exhibits a higher percentage of substituents (sulfate, methoxy, and pyruvate groups) at the C-2, C-4, and C-6 carbons of the β-d-galactose unit and at the C-2, C-3, and C-6 of the other sugar monomer [70]. Agarose is a polymer of the repeating unit [-d-galactose-β-(1→4)-3,6-anhydro-l-galactose-α-(1→3)-] and is mainly responsible for the agar gelling properties, while agaropectine is formed from the repeat unit [-d-galactose-β-(1→4)-l-galactose-α-(1→3)-] [70].

As other polysaccharides extracted from algae, agar also exhibits interesting biological activities that may soon lead to its application as a medicine. For example, agars can act as cytotoxic, hypoglycemic, and anticoagulant agents while its hydrolysis produces agaro-oligosaccharides with antioxidant, anti-inflammatory, and anti-α-glucosidase activity [49,66].

##### Carrageenans

Carrageenans are sulfate polysaccharides not assimilable in the human digestive tract and obtained mainly from red seaweed, most of them edible. These dietary fibers have a high commercial value, mainly because they can form thermoreversible gels with applications in the cosmetic (e.g., as emulsion stabilizers, film formers, and hair conditioning agents), pharmaceutical (e.g., in drug delivery, tissue engineering, and antivirals) and food industries (e.g., as E407 or E407a,, mainly in dairy products) [71,72,73].

Carrageenan content is species specific, depends on ecophysiological factors, and in the literature, values ranging from 22 to 88% (dw) can be found [49].

Carrageenans are linear chains composed by the repetition of the following sequence [-β-d-galactose-(1→4)-3,6-anhydro-α-d-galactose-(1→3)-]. In this base structure, if there is only one sulfate group at C-4 to the β-d-galactose unit, the polysaccharide is named kappa (κ)-carrageenan; if it accumulates a second sulfate group at C-2 of the 3,6-anhydro-α-d-galactose unit, it is named iota (ι)-carrageenan). Some species of red algae have a third type of carrageenan (lambda (λ)-carrageenan) composed only of galactose units (3,6-anhydro-α-galactose units are absent), where there are sulfate groups at C-2 and/or C-6, and with an alternating (1→3) and (1→4) glycosidic linkage [73,74]. Several other types of carrageenans (e.g., θ-, μ-, and β-carrageenans) are described in the literature [74], but their study and industrial applications are still not relevant. The chemical composition of carrageenans described above renders carrageenans soluble in water, where they form viscous solutions, and are stable at different pH values.

Carrageenans are mainly extracted from the *Eucheuma* and *Kappaphycus* species and to a lesser extent from the *Gigartina* and *Chondrus* species, but they have a diverse chemical composition. The *Kappaphycus alvarezii* and *Eucheuma denticulatum* species are the major sources of κ-carrageenan and ι-carrageenan, respectively, while κ-carrageenan and λ-carrageenan are obtained mixed mainly from the *Gigartina* and *Chondrus* species [51,73].

The distinct chemical composition of each carrageenan type implies different gelling properties. Thus, λ-carrageenan does not form gels while K^+^ promotes the gel formation of κ-carrageenan and ι-carrageenan gelling by the addition of Ca^2+^. Native carrageenans have the GRAS (generally recognized as safe) status, although there have been some studies reporting inflammatory reactions and some degree of toxicity to the colon microbiome at least in some niche of the population, most probably caused by degraded carrageenans [49,75]. From a pharmacological point of view, carrageenans or under its trade name carraguard, exhibit antiviral activity against swine fly virus subtype H1N1, human immunodeficiency virus (HIV), herpes simplex type 1 and 2, and the human papillomavirus (HPV) [52]. In fact, the carrageenans’ anti-HPV activity is the basis of the development of two ongoing clinical trial studies (ClinicalTrials.gov Identifier: NCT02382419 and NCT02354144). Carrageenans also showed interesting in vitro and in vivo cytotoxic, anticoagulant, anti-hypercholesteremic, antioxidant, and neuroprotective activities [76,77]. Despite the above activities, the use of carrageenans as an active ingredient in the pharmaceutical industry is not yet a significant reality. Nonetheless, carrageenans have been already included in the Britannica, European, and American pharmacopoeias, and therefore, will soon be used as pharmaceutical excipients.

##### Ulvans

Ulvans are the main family of sulfated water-soluble heteropolysaccharides with gelling properties extracted from green seaweeds, and whose glycosidic linkages are not degraded by both human enzymes and human colonic bacteria [78]. The ulvan content ranges from 8 to 29% (DW) of green algae [78] and are mainly composed of rhamnose and xylose sulfated together with glucuronic acid and small quantities of iduronic acids and glucose [49,73,78]. The sugar sequence in ulvans is difficult to determine, not only by the degradation caused by the extraction methods and its species variability, but mainly because they have a very heterogeneous sugar composition, several short repeating units (as discussed below) and branches, and, therefore, in opposition to other seaweed polysaccharides, creates a disordered complex 3D structure. However, some common sequences have already been identified. The most frequently reported repeating unit are: the A3s type [-α-l-rhamnose-3-sulfate-(1→4)-β-d-glucuronic acid] and the B3s type [-α-l-rhamnose-3-sulfate-(1→4)-α-l-iduronic acid]; the less frequent ones are the U3s type [-α-l-rhamnose-3-sulfate-(1→4)-β-d-xylose], U2′s3s type [α-l-rhamnose-3-sulfate-(1→4)-β-d-xylose-2-sulfate-], and the A2g3s type where the main chain of rhamnose and glucuronic acid (like on A3s type) arises branched at C-2 of rhamnose with a unit of glucuronic acid [73,78]. Although there are some data that seem to indicate the existence of small amounts of other sugar monomers in the ulvan structure, like galactose and glucose [79], these findings need clarification because it is not known how these units are incorporated [73].

Ulvans could be a source of rhamnose, the major component of ulvans, and a rare sugar in terrestrial plants, that can be used as a precursor for the synthesis of flavored compounds [49]. On the other hand, ulvans are a unique potential source of iduronic acid in plants, an acid that is used in the synthesis of anticoagulant analogues of heparin fragments [49]. Ulvans exhibit diverse biological activities such as immunomodulatory, antioxidant, and anticoagulant, although their activity depends on the sulfation degree, the three-dimensional structure of ulvans, and the proportion of sugar’s constituents [80,81,82,83]. Studies focused on the isolation and characterization of green algae polysaccharides continue and have been conducted to identify polysaccharides with new structures such as homo- and hetero-sulfated rhamnans [84,85].

##### Alginates

Alginates, also known as algin, are constituted of units of mannuronic and guluronic acids and are responsible, at least partially, for seaweeds’ flexibility. The presence of a carboxyl group in each monomer confers an anionic character that is variable with pH to the alginates, and because of this, it can be extracted in salt (called alginates) or acid (alginic acids) form. The polymer portions composed of guluronic acid (α-1→4) exhibit a coiled structure, forming cavities where Ca^2+^ ions or other ions like Fe^3+^ and Al^3+^ [86] can be housed, stabilizing the structure and conferring the gelling properties to the alginates by which they are so commercially known. When placed in water that absorbs up to 200–300 times its dry weight, alginates form a viscous gum with immense applications. The proportion of α-1→4 guluronic acid segments and the length of these segments define the properties of the alginates (a greater proportion of mannuronic acid units produces elastic gels whereas a greater proportion of guluronic units originates harder gels), properties that vary significantly from species to species and with the ion trapped [33,86]. For example, alginates from *Laminaria hyperborea* contain about 60% of α-l-guluronic acid segments, while other sources showed 14–31% of this structure type [87,88].

Alginate content is also specific (e.g., from 16 to 23% (dw) on Ghanaian *Sargassum* and *Padina* species [88]). Commercial alginates are extracted mainly from *Ascophyllum nodosum, Laminaria hyperborea, Saccharina japonica, Macrocystis pyrifera*, and *Laminaria digitata* where its content can reach 44% (dw) [89]. In addition, the alginate content varies with season and growth conditions (growth in turbulent zones increases the alginate content) [33].

Alginates are non-toxic hydrogels with excellent biocompatibility properties that promote their application in the pharmaceutical industry as auxiliaries in drug delivery, tissue healing, and relief from esophagitis and heartburn symptoms [87,88,89]; in the cosmetics industry as gelling agents; and in the food industry as a non-toxic additive (E400–405) [90]. Some studies have shown that alginates exhibit promising antitumor and antibacterial activities [50,51], opening the possibility for its use, soon, as the active ingredient in new drugs.

##### Laminaran

Laminaran is a storage homopolymer of β-d-glucose (β-glucan) linked by a 1→3 glycosidic linkage (and a very occasional 1→6 linkage that forms a branch, and simultaneously an interchain linkage), which may have a mannitol unit in their reducing end or a few uronic acid residues [91,92]. The polymerization degree is around 20 to 25 units, while the solubility of the laminaran depends on the branching degree (more branches correspond to great solubility) [93,94].

It is possible to find very different laminaran content values in the literature that range from, for example, 0.03% (dw) in *Fucus spiralis* [93] and 30% (dw) *in Laminaria hyperborea* [34]. As laminarans are polysaccharides of storage, it is produced after a phase of bigger seaweed growth, which means that its content is season specific and is highest at the beginning of winter [94]. However, even for the same species harvested in the same season, very different values have been reported in the literature such as the work published by Graiff et al. [93], which reported a very low content of laminarin for *Laminaria hyperborea* (0.86% dw), and justify that, although geographic differences should be considered, the difference to other higher values is mainly due to a large quantitative overestimation caused by the method used in previous studies.

Although laminarin lacks the ability to gel or to increase the viscosity of an aqueous solution, its use for medical and pharmaceutical purposes is of significant interest. In fact, some studies have shown its antitumor action, its potential as a safe surgical dusting powder, and the anticoagulant action of its sulfated form [33,34].

#### 3.1.2. Minerals

Although minerals are essential to the life cycle of macroalgae, they are not biosynthesized by them, but rather absorbed from the environment, depending on several factors such as salinity, temperature, pH, and light. Except for some cases, macroalgae have higher mineral content than terrestrial plants due to their excellent bioabsorption and bioaccumulation skills [51]. This can be confirmed by comparing the total mineral content, determined as ash content, as indicated in Table 1 (7 to 77% dw), and the values exhibited, for example, by spinach raw (15 to 26% dw) [91] (the most mineral-rich land vegetable), by *Aloe vera* (14.1–16.0%) [92], a raw material used in several cosmetic commercial products, or by mango kernel (1.5–3.2%), a source of an excellent oil also used in cosmetic products [95]. The analysis of Table 1 also shows that the total mineral content as well as the content of each mineral, varies substantially, depending on the phylum/species of macroalga, of the geographic zone where it grows, and of the harvesting season [38,39,49,96].

Essential minerals are found in seaweeds such as the most abundant minerals Na, K, Ca, Mg, and P (also called macrominerals), and trace elements like Zn, Se, Fe, Cu, Mn, and I. Table 1 shows that the total content of Na, K, Ca, and Mg ranges from 6.6 to 8.9% in green seaweeds; 9.7 to 20.7% in brown algae; and 7.8 to 8.9% in red algae. These values are similar to or higher than those found in the aerial parts of *Aloe vera* (6.7–7.8%) [92], or in other terrestrial plants such potato (6.0%) [97]. Although sodium content is higher in macroalgae than in many foods (e.g., spinach 0.91% [97]), in medicinal plants (*Salvia officinalis* 0.012% [97]), or in some cosmetic ingredients (e.g., *Aloe vera* 0.2–0.4% [92]), which may increase the risk of cardiovascular diseases and hypertension [98], the potassium content is also high in seaweeds. Thus, the ratio Na/K will not be as unfavorable to good health as one might think [99]. In several brown algae, this ratio is less than 1 [38,96], which will contribute to the maintenance of good cardiovascular health, but not as low as that found, for example, in Aloe vera, sage, or greater celandine (0.006–0.09%) [95,97,100].

The other three macro-minerals present in seaweeds are Ca, P, and Mg. Calcium plays a relevant role in the body skeleton, in the strength of the heart and smooth muscle contraction, and in the nervous and muscular equilibria [101], while magnesium is a very relevant cofactor of many enzymes such as those involved in cellular respiration [102]. With the exception of some red macroalgae, seaweeds exhibit a higher content of Ca than P (Ca/P >> 7, Table 1), which is identical to that found in *Aloe vera* and sage [92,97]. It should be noted that the Ca/P ratio is correlated with metabolic bone disease and the skeletal mineralization process [103].

Several elements like Fe, Mg, Zn, Cu, and Co are involved in many metabolic processes and also act as cofactors of enzymes, while I is an essential component of thyroid hormones, helping the body regulate metabolism, growth, and development and their deficiency may be the cause of many health problems such as cardiovascular, anemia, or hypothyroidism diseases [96,104]. Although there are variable contents depending on species and environmental factors, the iron content reaches higher values in the group of green algae [40,96] (Table 1), while zinc (mainly between 3–100 mg/kg) and copper (mainly between 0.5–17 mg/kg) are both trace elements with the lowest inter-group variation (in percentage terms).

The Mn content varies very significantly among species, with three algae groups with the highest content (red seaweed *Agarophytum vermiculophyllum* 392 mg/kg d.w. [36]; green algae *Ulva intestinalis* 180 mg/kg d.w. [40]; and brown seaweed *Fucus serratus* 150 mg/kg d.w. [36]).

The iodine content is very variable both between species and throughout the year, and values can be observed ranging from 0.0023% in green seaweed to 1.0–1.1% in brown seaweeds *Laminaria hyperborea* and *Laminaria digitata* [38,40], with the iodine content, on average, highest in brown algae [96] (Table 1).

Bromine is a relevant element in the development of tissues and in the construction of the basement membrane and has therefore been recently considered as an essential element [105]. Although there is little literature data on its content in macroalgae, it may range between 0.51–1.2, 0.26–0.65, and 0.64–0.83 g/kg (d.w.) in green, brown, and red seaweeds, respectively [36].

Minerals, as already mentioned, are essential to good health, and there are already in vivo studies showing that many minerals in macroalgae exist in forms that exhibit high bioavailability [106] such as calcium phosphate, which exhibits greater bioavailability than in other forms [107].

Despite the benefits of the minerals above-mentioned, it cannot be ignored that an excess of some of these minerals may also pose a health risk. In fact, for example, as algae are very rich in sodium and iodine, excessive consumption may contribute to the onset/aggravation of problems of hypertension and hyperthyroidism [96,108]. Furthermore, it should be accepted that, due to the high capacity of macroalgae to accumulate elements from the environment and to the increasing problems of marine pollution, undesirable elements such as arsenic, cadmium, mercury, and others can accumulate at such levels as to limit the use of these algae in applications as feed ingredients [40].

#### 3.1.3. Proteins

Proteins are, after carbohydrates and minerals, the third most abundant class of primary metabolites found in macroalgae [109]. These are polymers resulting from the combination of different amino acids linked by an amide bond, where there are different ratios of each amino acid and 20 different amino acids (Figure 1), nine of which are essential amino acids (i.e., that must come from food).

Proteins perform different functions in all living beings such as repair and maintenance, mechanical support, energy, and the transport and storage of molecules and ions, enzymes, antibodies, hormones, etc. Macroalgae are a valuable source of protein not only for their content, but mostly their composition of amino acids because they contain all nine essential amino acids (histidine, isoleucine, leucine, lysine, methionine, phenylalanine, threonine, tryptophan, and valine) [53].

It should be noted that the comparison of different protein contents from various studies is not always easy or feasible, considering the differences of the quantitative extraction procedures [109,110], or different ways of calculations (sum of each amino acid residue content or the use of a nitrogen-to-protein conversion factor). Above all, it is the use of different values for nitrogen-to-protein conversion (6.25 is mainly used for different types of food; 5 is recommended for seaweeds by Angell et al. [111]; or 4.59, 5.13, and 5.38, depending on the group to which the seaweed belongs [112]). Taking into account the studies in which this comparison is possible, it can be concluded that the protein content and its amino acid composition, as in the case of other primary metabolites, varies with the seaweed group and species [43]. In general, the lowest protein content is found in brown seaweeds and the highest in red seaweeds, while green algae exhibit a range of protein levels that partially overlap with the lowest levels in red algae [31,41,51,53]. Some species of the *Heterosiphonia* and *Porphyra* genus exhibit exceptional protein contents, reaching 33% and even 50% (d.w.), respectively [28,41,43,113], values similar to those found in soybean (38% d.w.) [97] and much higher than those found in samples of *Aloe vera* leaves (4.7–8.3% d.w.) [92] or corn meal (4.1% d.w.) [114].

The environmental, seasonal, and geographic conditions have a significant influence in the protein content of seaweeds [28,29,30,51]. For example, the protein content of *Saccharina latissima* collected in the same place can range from 1.3 to 10.8% (d.w.), while the percentage of glutamic plus aspartic acids goes from a minimum value of 19–26% total amino acids to the maximum percentage of 42–49% eight months later [115]. There are some indications that protein content shows a negative correlation with temperature and salinity [28,30].

Seaweeds are an excellent source of glutamic acid [51], this amino acid being the most abundant (9.5 to 19% of protein) in practically all 27 species of the three macroalgae groups analyzed by Mæhre et al. [114] and Astorga-España et al. [43], followed, in most cases, by aspartic acid with 8.2–12.6% of protein [43]. High levels of these two amino acids are responsible for the typical taste of marine products and for the umami flavor of macroalgae [43,53].

Generally, the amount of essential amino acids in proteins from the three seaweed groups is around 46% of the total amino acids [111], although some authors have reported lower values such as 33.5 to 42.3% of total amino acid [114], with the red algae presenting a higher percentage of essential amino acids, which means a similar or even higher protein quality when compared with fish protein derivatives or soybean [111].

Methionine and lysine are essential amino acids whose abundance in the diet is more limited, so they are often complemented artificially. In macroalgae, the methionine percentage, 2–3% in relation to the total content of amino acids, is higher than in soybean (1.2%), but slightly less than in fishmeal (3.1%), while on average, lysine exits at levels lower than those found in fishmeal [53]. However, there are some exceptions such as *Ballia* species where lysine is the second most abundant amino acid [43]. Tryptophan is also an essential amino acid whose percentage in relation to the total amino acid content is quite low, either in seaweeds and in food products such as soy or fishmeal (less than 2%), but which is particularly abundant in red algae where its content is more than double that of green algae [53]. However, it should be noted that a high percentage of essential amino acids does not mean a high concentration of these amino acids in the whole alga, as shown by Angell et al. [111], when compared to foods of high protein content. This is particularly true for species of algae with below-average protein levels and should be considered in applications where whole algae are used and not its protein extracted.

The seaweed proteins are also important as a source of peptides and amino acid extracts, mainly after enzymatic digest, which has the advantage of increasing their solubility in water and thus makes them suitable for use in various industrial applications [116]. Some of these peptides and protein extracts are already on the market to treat hypertension and obesity, while others can be incorporated into nutraceutical food [52,117]. There are still other peptides and protein extracts that exhibit adequate properties to be used in cosmetic formulations as antioxidants and/or as moisturizers [116,117]. For example, the peptide obtained from seaweed protein waste (Figure 1) exhibits a high antioxidant activity by different mechanisms [116], and the protein extracts and peptides of *Porphyra*, used in cosmetic formulations for greater skin and hair hydration and shine ability, its use patented in 2010 [118].

#### 3.1.4. Lipids

Lipids are a group of compounds made up of different subgroups such as fatty acids (saturated, mono-, di-, and poly-unsaturated), waxes, acylglycerols (mono-, di-, and triacylglycerols), phospholipids, glycoglycerolipids, fat soluble vitamins (e.g., vitamins A and E), sterols, and carotenoids (Figure 2).

Lipid content in macroalgae is considered low (0.3–5%, Table 1) when compared to other species such as the plant *Mangifera indica*, which is known to be the source of mango kernel oil (lipid content of 10–15%), an emollient used in various topical cosmetic preparations [95]. However, seaweed lipid content is similar to that found in *Aloe vera* (2.2–2.7% on dry basis), a species also used in cosmetic formulations [92]. Exceptionally, some macroalgae exhibit a high lipid content such as in the cases of Hawaiian coast brown seaweed *Dictyota* species (16.1–20.2% d.w.) [121], the tropical green seaweed *Caulerpa sertularioides* (13.04% DW) [122], and the Tunisian *Ulva lactuca* (7.9% d.w.) [44]. Lipid seaweeds are variable and dependent on factors such as species, geographical region, season, and environmental parameters like salinity, temperature, and light intensity [30,44,52]. Lipid content also depends on the reproductive state of seaweeds, for example, *Ulva* in its reproduction processes forms many mobile spores or gametes rich in lipids. This could be the reason for the high values in Tunisian *Ulva lact*uca. For example, the lipid content of *Sargassum muticum* collected in Vigo, Spain, could change from 1.6 to 3.2% (d.w.) throughout the year [30], while the same species *Ulva lactuca* L. collected in a different country exhibited a lipid content of 0.3–3.6% (d.w.) [44]. In general, green algae, on average, exhibit lower total lipid content while brown algae consistently exhibit values closer to the upper limit indicated above [34,37].

Aside from its low quantity, the chemical characterization of seaweed lipids has been the subject of much research, as shown below, since many of them are unique and play key roles in nutrition and cell membrane construction such as the essential fatty acid α-linolenic acid (ALA), which is not synthesized by mammalia and is biosynthesized by terrestrial plants in limited quantities [123].

The most abundant lipid class may be phospholipids or glycoglycerolipids [46,121]. On the other hand, in green seaweeds, acylglycerols exist in higher quantity than the sum of more polar classes (phospholipids and glycoglycerolipids), able to reach 50–70% of the total lipids [44,52], while in red seaweeds, the opposite is true, with polar lipids the most abundant [44,46]. The most relevant research in seaweed lipids is on the fatty acid profile, whether as free fatty acids (a very small amount), glycerolipids, glycoglycerolipids, or phospholipids, the last two being the most abundant. Determination of the fatty acid profile in macroalgae includes determining the content and carbon chain length of each fatty acid as well as, if any, the number and position of the C–C double bonds in the carbon chain. Seaweeds are a source of fatty acids with mainly 14 to 22 carbons, although the most abundant, also in this case, depends on which group belongs to seaweed. Thus, in brown and red algae, C20 fatty acids (mainly arachidonic and eicosapentaenoic acids) constitute the majority, while in green algae, C16 and C18 fatty acids are the most abundant [35,51]. It should be noted that the fatty acid profile of macroalgae is species specific, it varies with environmental and geographic factors [44,51], and even with the life cycle phase of the species [34].

Based on the number of C–C double bonds, the fatty acids can be classified as saturated (absence of double bonds), mono-unsaturated (one double bond only), or poly-unsaturated (two or more double bonds). The fatty acid profile in red and green seaweeds shows that saturated fatty acids are predominant [29], while brown seaweed has a majority of unsaturated acids [31]. The most abundant saturated fatty acid in seaweeds is palmitic acid (C16:0) [29].

The position of the first double bond counted from methyl *terminus* allows the unsaturated fatty acids to be classified into omega-3 (n-3), omega-6 (n-6), or omega-9 (n-9). The essential poly-unsaturated fatty acid octadecatrienoic acid (also named linolenic acid, ALA, C18:3) is the precursor of eicosapentaenoic acid (EPA, C20:5), docosahexaenoic acid (DHA, C22:6) and docosapentaenoic acid (DPA, C22:5), and they are all the most relevant omega-3 seaweed fatty acids, while arachidonic acid (C20:4) and octadecadienoic acid (also named linoleic acid, C18:2) are the major omega-6 fatty acids of seaweed. In fact, seaweeds accumulate high levels of these poly-unsaturated fatty acids, which have beneficial effects to human health, contributing to cardiovascular risk reduction and also improving both the immune response and brain function [30,52,124]. In general, seaweeds have a well-balanced fatty acid composition, once they exhibited a lower omega-6/omega-3 ratio lower than 10, the ratio recommended by the WHO; lower than 5 recommended by the European Nutritional Societies [51]; or even between 0.5 to 1 as suggested by Simopoulos and DiNicolantonio [125], and are considered as having the potential to reduce the risk of several diseases such as cardiovascular, inflammatory, and neurological. Red seaweeds exhibit mainly an omega-6/omega-3 ratio ranging from 0.1 to 3.5 [31,34,40,48], green seaweeds exhibit a ratio from 0.3 to 1.7 [31,34,52], while most brown seaweeds have shown an omega-6/omega-3 ratio less than 1 [31,34,40,121]. In comparison, in *Aloe vera* leaves, the content of the omega-3 linolenic acid (ALA) is higher than that of the omega-6 linoleic acid [126], while in mango kernel oil and palm oil, it is the opposite [127].

However, as already shown by several authors [30,34], the temperature of the water in which the seaweed grows determines, on a large scale, both the content of polyunsaturated acids and the number of unsaturated fatty acids, which are both higher in algae from cold waters.

The high content of unsaturated fatty acids in macroalgal lipids may present a potential problem since their degradation by oxidation reactions may occur on a large scale. However, several studies reviewed by Miyashita et al. [121] showed that poly-unsaturated acids of macroalgae existing mainly as glycoglycerolipids exhibit high oxidative stability, superior to that exhibited by phospholipids or triglycerides from the oil of salmon roe or sardine.

Besides fatty acids, seaweed lipids include pigments, lipophilic compounds also named carotenoids (xanthophylls and carotenes), which differ according to the group of algae: in red seaweeds, the most common are β–carotene, zeaxanthin, and lutein; violaxanthin and neoxanthin in addition to β–carotene and zeaxanthin in green seaweeds, while brown algae are rich in fucoxanthin [35,128]. The brown algae *Sargassum horneri* can accumulate up to 5.6% of fucoxanthin (total lipid basis) [129]. Fucoxanthin has great commercial value since it is used in obesity treatment, helps to regulate dyslipidemia and reduce the risk of cardiovascular diseases, and exhibits antioxidant and antitumor activities [34,52,130].

Sterols, because of their lipophilicity, are also a group of compounds included in the lipid class. Fucosterol is the most abundant sterol in brown algae [129] and can reach values of 0.7–2.3 g/kg (d.w.) in some *Undaria* and *Laminaria* species, showing that this sterol has several health benefits including antioxidant, antiphotoaging, anti-atopic, and anti-inflammatory effects [51]. The brown algae *Sargassum horneri* can accumulate up to 16.2% of fucosterol (total lipids basis) [129]. In red seaweeds, cholesterol is the major sterol [44], while green algae contain mainly isofucosterol and cholesterol [44]. The seaweeds also contain a very specific type of sterol in low concentration, but whose pharmacological activities have aroused much interest, for example, the sulfated steroids that exhibit high anticancer and antibacterial effect even against resistant strains [131]. Capisterones A and B isolated from green seaweed *Penicillus capitatus* enhanced fluconazole activity in resistant strains [132].

Both carotenoids and sterols are lipophilic compounds that are biosynthesized via coupling of isoprene units, the same pathway to the secondary metabolites diterpenes, but are very different from those of the other lipids, and therefore carotenoids and sterols are often included in secondary metabolite classes.

Green, brown, and red seaweeds also contain other lipophilic compounds such as vitamin E (α-tocopherol) [34,44], with one of the highest contents found in brown seaweed *Sargassum wightii* and in red seaweed *Gelidiella acerosa* (1300 mg/kg d.w.) [37], while green algae *Ulva lactuca* only accumulate 0.03 mg/kg (d.w.) [133]. Seaweeds also contain fat-soluble vitamin A (3.4 and 4.0 mg/kg d.w. on *Gelidiella aceros*a and *Sargassum wightii*, respectively [37]) and compounds that industrially can be converted into vitamins E and K such as phytol [44]. Furthermore, green and red algae contain significant quantities of the non-toxic natural coloring β-carotene, a provitamin A compound, which can accumulate up to 4.5 g/kg (d.w.) in *Porphyra* sp. and 0.3 g/kg (d.w.) in *Ulva* sp. [34].

As above-mentioned, the lipid class is, among the classes of primary metabolites, the least abundant in macroalgae. However, it is the one that exhibits the greatest structural heterogeneity, and the one where the line separating the classification of metabolites as primary or secondary is more diffuse.

### 3.2. Macroalgae Secondary Metabolites Diversity

In 1977, D. J. Faulkner published a review where he emphasized the unusual molecules found in marine organisms [134]. This article prompted the author to start a series of reviews (19 reviews published in Natural Product Reports from 1984 to 2002) describing the reporting of marine natural products including those isolated from microalgae and, where, for the first time, he used the above-mentioned macroalgae groups [135]. From these reports, it is possible to detect that macroalgae are a rich source of diverse secondary metabolites, and more than 440 compounds are described by Faulkner, with the red macroalgae providing more compounds (>230) and the green macroalgae having fewer compounds (~30). In his last report [136], although fewer compounds isolated from macroalgae are referred to (~55), the tendency for the greater richness of the red algae in secondary metabolites was also observed. In 2003, Blunt et al. [137] picked up the series, published an actualization every year, and surprisingly in their last publication, they reported 67 new secondary metabolites isolated from macroalgae [138]. In a review published in 2010, the authors showed an interesting graphical analysis from which the number of secondary metabolites isolated from each macroalgae type per year can be seen [139]. Considering their following publications, it is possible to complete the data, and as seen in Figure 3, in the last 11 years, more new compounds have been discovered, nearly twice the number reported in the first 40 years. It is also possible to confirm that red macroalgae still provide more new compounds whereas the green ones give fewer compounds (Figure 3). It is clear that the red and brown macroalgae have attracted more attention from the scientific community most likely because there are more species of these macroalgae [140]. Additionally, it can be noticed in the three groups that in the last years, fewer compounds have been isolated (Figure 3). Despite the recent shift away from macroalgal phytochemistry, the interest in their secondary metabolites is still high due to their vast biological applications [52,141]. In the following sections, the characteristic secondary metabolites will be presented, highlighting their biological value. It should be stressed that the aim was not a complete review of all metabolites, but an author’s selection of the most significant compounds, bearing in mind their medicinal and/or cosmetic applications as well as their relationship with the macroalgae group.

#### 3.2.1. Chlorophyta Secondary Metabolites

Chlorophyta are the group of macroalgae least prolific in secondary metabolites (Figure 3) apparently due to their rapid growth and/or their environment, as the surrounding environments appear to have an impact on the genetic expression of the secondary metabolites [135]. These macroalgae are known to produce secondary metabolites bearing the 1,4-diacetoxybutadiene moiety (Figure 4), being the first reported examples flexilin **12**, a compound isolated from *Caulerpa flexilis*, and trifarin **13** (Figure 4), a compound isolated from *Caulerpa trifaria* [149]. In the same year, the isolation from *Caulerpa prolifera* of another sesquiterpenoid, named caulerpenyne **14**, bearing the same moiety was reported [150]. A year later, the isolation and biological evaluation of the sesquiterpenoid rhipocephalin 15 (Figure 4) from the green alga *Rhipocephalus phoenix* was reported [151]. Our interest in highlighting this derivative is due to the fact that the authors also demonstrated its potential to protect the alga against predators. This biological behavior was demonstrated by other derivatives, for example, udoteal 16 (Figure 4), isolated from *Udotea flabellum* [152], and it is anticipated that the 1,4-diacetoxybutadiene moiety may be regarded as a chemical defense produced by green alga [153,154]. In the subsequent years, several other derivatives bearing this important moiety were found with some less common structures such as the bicyclic diterpene **17**, isolated from *Caulerpa trifaria* [155]; diterpenoid **18**, isolated from *Pseudochlorodesmis furcellata*; and the diastereoisomeric mixture **19**, which is an oxidized form of caulerpenyne **14**, isolated from *Caulerpa taxifolia* [156], just to mention a few examples (Figure 4). These enolate esters are, in biological systems, hydrolyzed to aldehydes so are often isolated in their aldehyde form. Moreover, these aldehydes show significant biological activities including ichthyotoxic. A significant example is halimedatrial **20** (Figure 4), isolated from *Halimeda* species such as *H. tuna, H. opuntia, H. monile, H. incrassata, H. cylindracea, H. simulans, H. gigas, H. scabra*, and *H. copiosa* [157,158]. However, diterpenoid **21**, found only in the species *H. tuna* and *H. scabra*, is also an interesting example [158]. Furthermore, the genus *Udotea* also revealed the presence of these biologically active aldehydes; the monocyclic diterpenoid **22**, isolated from *Udotea flabellum*, is also interesting from a biological potential point of view [159].

Although not as prevalent as the above-mentioned, other terpenoid motifs have been found in green algae species, and several have been discussed in the primary metabolites section such as carotenoids and sterols. Here we can emphasize, for example, that the norcycloartene triterpenoid 23 (Figure 4), isolated from *Tydemania expeditionis*, is among the first reported derivatives [160].

In our selection of the more noteworthy secondary metabolites, the squalene derivatives cannot be ignored, not because they are widespread in green alga, but because their occurrence justify the seaweeds’ use in cosmetics [117]. In that context, it makes sense to emphasize the isolation of (S)-(-) squalene-2,3-epoxide 24 (Figure 4) from *Caulerpa prolifera* because it was the first time that it was found in a natural source [161], although other derivatives were also found later on [162].

Green algae are also recognized for their richness in unusual polypeptides such as kahalalides [163] from which kahalalide F **25** (Figure 4) was, as far as we are aware, the first member to be reported [164]. Moreover, its biological potential was also demonstrated and has prompted other authors to develop an efficient synthetic methodology and confirm its structure [165] including its stereochemistry [166,167]. Other new polypeptides have been found, mainly in the *Bryopsis* species [168,169], but not only, for example, interesting derivatives have also isolated been from *Derbesia marina* [170].

Although halogenated compounds are not commonly found in green algae [171], some examples of the isolation of these metabolites can be found in the literature, and they represent our final choice of green algae metabolites. To the best of our knowledge, the first report was the isolation of some brominated monoterpene quinols from *Cymopolia barbata* [172], but several other examples have been isolated from the same alga [173,174,175]. Interesting examples have also been found in *Avrainvillea* species [176,177], from which rawsonol **26** (Figure 4) can be highlighted due to its ability to inhibit the HMG-CoA reductase [177].

#### 3.2.2. Ochrophyta Secondary Metabolites

Brown algae are richer in secondary metabolites than green algae (Figure 3) and phlorotannins, phenolic compounds, are their characteristic secondary metabolites [178,179]. Other important compounds include several types of terpenoids and small acetogenins [138].

The exclusive brown algae phlorotannins are phloroglucinol **27** (Figure 5) polymers that can be divided, accordingly with the linkages between the phloroglucinol units, into subclasses. Some authors have proposed six subclasses: phlorethols, fucols, fucophlorethols, eckols, fuhalols, and isofuhalols [180]; others only recommend the first four and suggest that fuhalols and carmalols are less common [181]. The most common linkage between the phloroglucinol units is ether bonds (e.g., trifuhalol **28**, Figure 5), however, carbon–carbon linkages can also be found, for example, in trifucol **29** (Figure 5). These phlorotannins also exhibit a broad range in size and the small ones showed interesting antioxidant activity [182], which gives value to the use of brown algae. Furthermore, phlorotannins are associated with UV protection, and consequently responsible for the use of some brown algae in cosmetics [117,183].

Other representative compounds are the diterpenes isolated from species of the genus Dictyota, which is one of the most studied genera and are richer in terpenes [184]. Earlier examples such as dictyol E **30** and amijiol **31** (Figure 5) and the more recent ones such as dictyoxide **32** and 4α-hydroxypachylactone **33** (Figure 5) are among the several bioactive diterpenes found in the Dictyota genus [185].

Brown algae are also recognized for their richness in important secondary metabolites such as fucosterol **10** and fucoxanthin **11** (Figure 2), which, due to their lipidic characteristics, were discussed among the other lipids (Section 3.1.4).

In our final examples, we would like to emphasize the secondary metabolites isolated from the genus *Cystoseira*, which is amongst the brown algal order Fucales, the genus that accounts for more reported structures. It is a genus rich in bioactive metabolites from which meroterpenoid derivatives can be highlighted [186]. In fact, these compounds are interesting from the biological activity point of view, but also due to their intramolecular transformations, which lead to other secondary metabolites [186]. A fascinating example is the case of the tetraprenyltoluquinol derivatives **34** and **35** (Figure 5).

#### 3.2.3. Rhodophyta Secondary Metabolites

Rhodophyta, as shown in Figure 3, are among the macroalgae that have provided, so far, more secondary metabolites both in diversity and abundance [138]. All the major classes of secondary metabolites are present in red algae naturally, except for the phlorotannins, which are exclusively produced by brown algae.

As above-mentioned, macroalgae can be distinguished through the analysis of their chemical profile [187]. However, what truly distinguishes red algae from the others are the halogenated secondary metabolites [171], of which more than 90% of the reported secondary metabolites contain bromine or chlorine [187]. These halogenated compounds are significantly associated with the ecological effects of these algae including chemical defense against epiphytic bacteria [188]. Different species of the family Bonnemaisoniaceae are well known as sources of halogenated compounds with strong antifungal and antibiotic activity [189].

Most of the compounds reported come from the Rhodomelaceae family, of which the genus *Rhodomela* and *Laurencia* have had more compounds isolated [171]. The *Laurencia* genus can, however, be emphasized due to the biological properties of their secondary metabolites [190], mainly sesquiterpenes and C15-acetogenins [191]. The sesquiterpenes isolated from this genus are typically cyclized, with laurinterol **36** [192] and pacifenol **37** [193] earlier representative examples (Figure 6).

Another interesting example is elatol **38** (Figure 6) due to its spiro-ring fusion less common structure, but also due to its potential as a lead compound in the development of new anti-leishmanial drugs [194]. Other terpenoid also found are diterpenes such as irieol A **39** [195] and triterpenes such as callicladol **40** [196] (Figure 6). Several other examples as well as their biological potential were recently reviewed [171].

The C15 carbon backbone, apparently originating from a C16 carboxylic acid through decarboxylation, leads to C15-acetogenin derivatives, which are a unique class of metabolites characteristic of the genus *Laurencia* [197]. Laurepinnacin **41** [198] and laurallene **42** [199] (Figure 6) are interesting examples, in fact, the laurallene **42** allene moiety is a challenge and attracted its total synthesis [200].

Other *Rhodophyta* families and/or genus produce halogenated terpenes and phenols, some are often highly halogenated [171], and halomon **43** (Figure 6) can be regarded as a notable example, not only due to the number of halogen atoms, but also because its antitumor properties have been intensely studied [201].

Peyssonoic acids A **44** and B **45** (Figure 6) are interesting secondary metabolites, isolated from red algae species belonging to the genus *Peyssonnelia* [202]. Their structure is somehow less common and the authors suspect that they may be responsible for macroalgae chemical defense against microbial attack.

*Plocamium cartilagineum* is a species of red algae (family Plocamiaceae, order Gigartinales). This species is characterized by its interesting secondary metabolites with a rich source of diverse polyhalogenated monoterpenes and a surprising degree of halogen incorporation [203].

Our final examples are the fimbrolides, for example, **46** (Figure 6), which has been mostly isolated from *Delisea pulchra* [204]. These halogenated furanones are suspected to function as intracellular signal antagonists [205] and to provide an antifouling defense [206].

## 4. Seaweeds Bioactive Effects: From Pre-Clinical to Clinical Studies

### 4.1. Seaweeds: In Vitro Bioactive Effects

Bioactive compounds isolated from seaweeds have been reported to possess a wide number of pharmacological in vitro activities (Figure 7). Thus, this section will mainly focus on the results of recent studies about the medicinal and pharmaceutical in vitro value of seaweeds.

#### 4.1.1. Antimicrobial Activity

Many seaweeds possess numerous bioactive components with antimicrobial properties [21]. Polysaccharides including carrageenans, alginates, agar, ulvans, fucans, laminarians, and their derivatives have been reported to be the main active compounds that show antiviral activity against viruses that are responsible for human infection [207].

Carrageenans extracted from red seaweed species including *Kappaphycus, Gigartina, Eucheuma, Chondrus*, and *Hypnea* [208], and their low molecular weight derivatives are shown to inhibit human herpes virus 1 (HSV-1) and 2 (HSV-2) in vitro [209]. It was reported that this inhibitory effect is mediated by blocking viral attachment and is influenced by the molecular weight and sulfation status [210]. Yamada et al. [211] reported that sulfation and depolymerization of carrageenan polysaccharides increased their anti-HIV activity. Galactan sulfate and xylomannan sulfate isolated from the red seaweeds *Aghardhiella tenera* and *Nothogenia fastigiate*, respectively, showed antiviral activity against a wide number of infectious viruses including human immunodeficiency virus (HIV), HSV types 1 and 2, and respiratory syncytial virus [212]. Agaroids, a high molecular weight galactan sulfate isolated from *Gracilaria corticate*, show antiviral property against HSV-1 and -2 by inhibiting initial virus attachment to the host cell [213]. Similar results were found with water-soluble sulfated polysaccharides isolated from two red algae *Sphaerococcus coronopifolius* and *Boergeseniella thuyoides*. These sulfated polysaccharides inhibited the in vitro replication of HIV and HSV-1 at 12.5 and 17.2 μg/mL, respectively [214]. Cimino et al. [215] reported that rosacelose isolated from the marine sponge *Mixylla rosacea* possess in vitro antiviral effects against HIV. The extracts of brown seaweed species *Hydroclathrus clathratus* and *Lobophora variegate* showed a potent anti-HSV activity against HSV-1 and 2 and a moderate anti-RSV activity [216]. Fucans, high molecular weight sulfated polysaccharides distributed in the cell walls of brown algae, are reported to inhibit the growth of various enveloped viruses [217]. Sulfated fucans isolated from *Dictyota mertensii, Lobophora variegata, Fucus vesiculosus*, and *Spatoglossum schroederi* could prevent HIV infection by blocking the activity of reverse transcriptase [218]. New fucose polysaccharides isolated from marine brown algae species, *Sargassum piluliferum*, exhibited an effective anti-influenza activity [219]. Additionally, a sulfated fucans extracted from Cystoseira indica demonstrated a promising activity against HSV-1 and -2 by inhibiting virus adsorption [220]. Furthermore, fucoidan, which is a polysaccharide based mainly on sulfated l-fucose and isolated from several species including *Adenocytis utricularis, Undaria pinnatifida, Cystoseira indica, Undaria pinnatifida* and *Stoechospermum marginatum*, possesses various antiviral activities [217]. It is effective against many RNA and DNA viruses including HSV1-2, HIV, dengue virus, and cytomegalovirus [221].

Halogenated compounds, sterols, heterocyclic, and phenolic compounds show antibacterial effects against both Gram-positive and negative bacteria [5]. Elatol, a halogenated compound extracted from red algae *Laurencia majuscula*, showed a significant antibacterial effect against *Staphylococcus epidermidis, Klebsiella pneumoniae*, and *Salmonella* sp. [222]. A halogenated furanone also known as fimbrolide isolated from *Delisea pulchra* was effective in the treatment of chronic *Pseudomonas aeruginosa* infection by inhibiting the furanone on the quorum sensing mechanism [205].

Laurinterol and *allo*-laurinterol, halogenated metabolites from the red alga *Laurencia* species, displayed a wide spectrum of antibacterial activity against methicillin-resistant *Staphylococcus aureus, penicillin-resistant Streptococus pneumoniae, vancomycin-resistant Enterococcus faecalis*, and *E. faecium* [223]. Lanosol enol ether originally isolated from the brown alga *Osmundaria serrata* exhibited an antibacterial and antifungal activity with a mean bacteriostatic and fungistatic activity of 0.27 ± 0.07 mg mL^−1^ and a mean bactericidal and fungicidal activity of 0.69 ± 0.15 mg mL^−1^ [224]. Five microalgal strains including *Pavlova lutheri, Chlorella marina, Nannochloropsis oculata, Dunaliella salina,* and *Isochrysis galbana* were tested for antibacterial activities by Srinivasakumar and Rajashekhar [225]. Overall inhibition of ethanol and petroleum ether extract varied from 13.17% to 16.22% against both Gram-positive and Gram-negative bacteria [225]. Priyadharshini et al. [226] tested the in vitro antimicrobial of methanol, butanol, and aqueous extracts of the marine macroalgae *Ulva fasciata*. A maximum inhibition zone diameter of 16 mm was detected against *Vibrio alginolyticus*, while a minimum inhibition zone diameter of 12 mm was detected against *Enterobacter* sp. The minimum anti-staphylococcal inhibitory concentration of five new C15 eight-membered cyclic ethers isolated from the red alga *Laurencia glandulifera* ranged from 8 to 256 μg/mL [227].

Several studies have confirmed that new therapeutics for acne vulgaris could be derived from algal extracts. β-d-galactosyl O-linked glycolipid found in the ethyl acetate extract of Fucus evanescens exhibited strong antibacterial activity against *Propionibacterium acnes* [228]. Similar results were confirmed for fucofuroeckol-A found in the *Eisenia bicyclis* crude methanolic extract, with a minimum inhibitory concentration (MIC) value ranging from 32 to 128 µg/mL [229].

Oral microbes are another important target of algal extracts. *Ulva linza* extracts have demonstrated antibacterial activity against *P. gingivalis* and *P. intermedia* [230]. Active compounds extracted from the green seaweed *Ulva linza* demonstrated effective anti-bacterial activities against *P. intermedia* and *Porphyromonas gingivalis* with MIC values of 39.06 µg/mL and 9.76 µg/mL, respectively [231]. Kim et al. [232] evaluated the anti-bacterial activity of ethanol extracts of *Saccharina japonica against Actinomyces naeslundii, Actinomyces odontolyticus, Fusobacterium nucleatum*, and *P. gingivalis*. The MICs ranged from 62.5 to 250 µg/mL and the MBCs ranged from 250 to 500 µg/mL [232]. Fucoidan was also found to be effective against oral bacteria either alone or combined with synthetic antibiotics with MICs ranging from 0.125–0.50 mg/mL and minimum bactericidal concentration (MBC) values ranging from 0.25–1.00 mg/mL [233].

Incidents of food poisoning due to microbial food spoilage have increased dramatically in the last few decades. Replacement of synthetic additives with natural antibacterial substances represents a new challenge for researchers. Different studies have been performed to confirm the antimicrobial action of seaweed extracts against food spoilage microorganisms and several edible antimicrobial films from these extracts have been developed and used as active food packaging. The methanol extract of the brown seaweed *Himanthalia elongata* inhibited *Listeria monocytogenes* by 98.7% at a concentration of 60 mg/mL [234]. Dussault et al. evaluated the anti-bacterial activity of the brown species, Padina and Dictyota. Results showed that their methanol extracts inhibited the growth of Gram-positive foodborne pathogens including *Listeria monocytogenes, Staphylococcus aureus*, and *Bacillus cereus* at concentrations lower than 500 μg/mL with no effect against Gram-negative species [235]. *Cladophora rupestris* extracts and *Scytosiphon lomentaria* methanol extracts were found to inhibit the Gram-negative bacteria *Salmonella typhimurium* [236]. Boisvert et al. reported that the extract of *Ulva lactuca* was a potent antibacterial against *Escherichia coli* while *Ascophyllum nodosum* was more effective against *Microccocus luteus* and *B. thermosphacta* [237]. A new active biodegradable film was developed based on polylactic acid and the brown seaweed *Fucus spiralis*; this was revealed to be effective against a wide range of microorganisms such as *Staphylococcus aureus, Pseudomonas fluorescens, Escherichia coli, Bacillus cereus, Bacillus subtilis, Aeromonas hydrophila, Vibrio alginolyticus*, and *Vibrio parahaemolyticus* [238].

#### 4.1.2. Antioxidant Activity

*Sargassum dentifolium, Laurencia papillosa, Ecklonia cava, Gelidiella acerosa, Hizikia fusiformis,* and *Jania corniculata* demonstrated a free radical scavenging activity and lipid peroxidation inhibition [239,240,241,242]. *Sargassum* sp. was also proven to inhibit glutathione-transferase activity [243]. The percentage of free radical scavenging activity for *Ulva fasciata* and *Chaetomorpha antennina* was about 83.95% and 63.77%, respectively [244]. This antioxidant activity is affected by location and salinity and is related to the presence of several compounds including ascorbate, vitamin E, carotenoids (fucoxanthin), sulfated polysaccharides, polyphenols (phlorotannins), carnosine, and glutathione [245]. Dieckol isolated from the brown alga *Ecklonia cava* and diphlorethohydroxycarmalol isolated from *Ishige okamurae* reduce the ROS induced by UV-B radiation in a dose-dependent manner and have potential whitening effects [246,247]. Sulfated polysaccharides from different sources such as *Sacccharina japonica* and *Ulva pertusa* have been demonstrated to possess antioxidant activity [248,249]. *Porphyra, Ulva, Ascophyllum,* and *Fucus* contain large amounts of vitamin E [34]. Phenolic compounds from the red seaweed *Kappaphycus alvarezii* were proven to exhibit scavenging and metal ion-chelating abilities [250]. Bromophenols from red alga *Polysiphonia urceolata* had a radical scavenging activity at an IC50 of 9.67 μM [251].

#### 4.1.3. Anti-Inflammatory Activity

An increasing number of studies have reported that bioactive compounds found in seaweeds could have possible anti-inflammatory activities. Different metabolites with anti-inflammatory properties were isolated from seaweeds such as phlorotannins, polyphenols, glycosterols, polysaccharides, and polyunsaturated fatty acids (PUFAs) [5,252]. Several mechanisms could be involved in these anti-inflammatory properties such as the inhibition of toll-like receptors, phospholipase A2, cyclooxygenase and nitrite formation; and the increase of anti-inflammatory cytokines including IL-6r and IL-10 [252,253]. The brown seaweed *Undaria pinnatifida* was observed to inhibit inflammatory response due to its high PUFAs content [254]. The methanolic extract of *Gelidiella acerosa* was found to be potent NO scavengers at 100 μg/mL [242]. Similar results were also reported for fucoxanthin isolated from brown algae, which was found to inhibit the production of NO in lipopolysaccharide-induced RAW 264.7 macrophage cells [255]. Agaro-oligosaccharides have also been revealed to suppress TNF-α production, a pro-inflammatory cytokine [256]. Fucoidan has potential anti-inflammatory effects through the inhibition of NF-κB, MAPK, and Akt activation in lipopolysaccharide-induced BV2 microglia cells [257,258]. Fucoxanthin, which is usually found in marine brown seaweeds, downregulate cytokine expression and reduce the inducible nitric oxide synthase (iNOS) and COX-2 mRNA over-expression in RAW264.7 macrophage-like cells as well as MCP-1 and IL-6 mRNA over-expression in differentiating 3T3-F442A adipocytes [259]. Dieckol obtained from *Ecklonia cava* was reported to decrease the over-expression of NF-κB, COX-2, and iNOS [260]. The ethanolic extract of brown algae *Ishige okamurae* was able to inactivate NF-κB in macrophages of RAW 264.7 cells [261]. *Sacccharina japonica* generated anti-inflammatory properties by decreasing the production of NF-κB, phosphorylation, and NO production in macrophages [262].

#### 4.1.4. Antiproliferative and Anti-Angiogenesis Activity

Antiproliferation and anti-angiogenesis activities of seaweeds have been evaluated in many in vitro studies and related to their antioxidant and immune modulator properties [263,264]. The methanolic extract of brown seaweed *Sargassum muticum* was found to be cytotoxic against breast cancer cell lines. This antiproliferative effect was dose-dependent and correlated with the total phenolic contents with an IC50 of 22 μg/mL and 55 μg/mL against MCF-7 and MDA-MB-231 human breast-cancer cell lines, respectively [265]. Polysaccharides extracted from *Sargassum pallidum* were reported to have significantly higher antitumor effect against HepG2 cells, A549 cells, and MGC-803 cells [266]. Fucans from the brown alga *Lobophora variegata* at 25 μg/mL decreased the cellular viability of tumor cell line HepG2 [267]. The anti-tumor effect of fucoidan is structure dependent and could widely be enhanced by lowering the molecular weight [268]. In vitro assays showed that the edible red seaweed *Eucheuma cottonii* has an antiproliferative effect against MCF-7 at 20 μg/mL and MB-MDA-231 at 42 μg/mL due to their high phenolic content [264].

#### 4.1.5. Anticoagulant Activity

Several studies have reported structural similarities between heparin and sulfated seaweed polysaccharides. As a rule, the anticoagulant activity is mainly attributed to thrombin inhibition due to the interaction between the negatively charged sulfated clusters with cationic proteins and is directly related to the polysaccharides’ molecular size and sulfate content [210]. Homogalactans from red seaweed such as *Gigartina skottsbergii*; and α-l-fucose-containing sulfated homo and heteropolysaccharides (fucan and fucoidan) from brown seaweed such as *Padina gymnospora, Dictyota menstrualis, Sagassum stenophyllum*, and *Spatoglossum schroederi* are known to possess anticoagulant activities [269]. The same activities are reported for the sulfated homo and heteropolysaccharides from green algae such as *Codium cylindricum, Caulerpa cupressoides*, and *Caulerpa racemosa* [210]. Athukorala et al. proved that the extract of *Codium fragile* and *Sargassum horneri* exhibited a potent anticoagulant activity due to their high content of high molecular weight polysaccharides [270]. A sulfated polysaccharide with a molecular mass between 8 and 20 kDa was isolated from the fermented brown seaweed *Sargassum fulvellum* and was proven to have an anticoagulant activity weaker than heparin [271].

### 4.2. Seaweeds: In Vivo Bioactive Effects

Several in vivo pharmacological effects were reported for bioactive compounds found in seaweeds (Table 3).

The methanolic extract of the brown seaweed *Spatoglossum schroederi* rich in flavonoids significantly inhibited the paw edema induced by carrageenan or dextran in mice. Results indicated that the anti-inflammatory properties were related to the inhibition of the release of inflammatory mediators and neutrophil infiltration; increasing the release of IL-10 [272]. Similar results were reported for the methanolic extracts of the brown seaweeds *Padina tetrastomatica, Dictyota menstrualis,* and *Sargassum wightii*; the red seaweed *Gracilaria edulis*; and the green seaweed *Caulerpa racemosa* [273,274]. The topical application of the methanolic red seaweed *Dichotomaria obtusata* inhibited mouse ear edema in a dose-dependent manner at an ED50 of 4.87 μg/ear, while the oral route showed low efficacy even at high doses due to the limited bioavailability of the active constituents [255]. Immune-suppressant effects by blocking the Th2 activity were reported for *U. pinnatifida, Porphyridium sp., Phaeodactylum sp.,* and *Chlorella stigmatophora* [301]. Ex vivo studies have revealed that fucoidan was able to protect human skin elastic fibers by reducing human leukocyte elastase activity [302]. Additionally, fucoidan has been found to help in the recovery process of immunologic function in irradiated rats by acting directly on macrophages, T lymphocytes, B cells, and natural killer cells [279]. Furthermore, it was reported that the antioxidant effects of fucoidan could protect against dopaminergic neuron death in the MPTP-induced animal model of Parkinsonism in C57/BL mice [280]. Monounsaturated fatty acid derivatives isolated from the green seaweed *Ulva lactuca* revealed cytoprotective properties by inducing many ARE-driven antioxidant genes in various mouse tissues in vivo [275]. Sulfated galactans isolated from the red seaweed *Chondrus ocellatus* showed antitumor and immunomodulation activities in mice of transplanted S180 and H22 tumor [283]. Fucoidan from *Undaria pinnatifida* and *Fucus vesiculosus* improved the activity of tamoxifen in both MCF-7 and ZR-75D breast cancer mouse models. The combination of tamoxifen with fucoidan from *Fucus vesiculosus* in the TOV-112d ovarian cancer mouse model had improved activity with no differences observed with the combination of tamoxifen with fucoidan from *Undaria pinnatifida* in an ovarian cancer mouse model [281]. Additionally, fucoidan from *Fucus vesiculosus* reduced HeLa human cervical tumor growth [282]. No changes in the activity of paclitaxel were observed when given in combination with fucoidan in either human ovarian cancer or breast cancer models [281]. Seaweed complex preparation also showed antitumor, radical scavenging, and immunomodulatory activities in transplanted hepatic tumor cell line H22 [303]. Fucosterol isolated from the marine algae *Pelvetia siliquosa* and dieckol were also found to reduce oxidative stress in vivo by significantly increasing the free radical scavenging enzyme activities and the activity of hepatic antioxidant enzymes such as glutathione peroxidase and superoxide dismutase [276,277]. Similar anti-cancer and neuroprotective activities have been reported for phytols present in the marine red alga *Gelidiella acerosa* [284,304].

*Halimeda incrassata* aqueous extract exhibited antioxidant and neuroprotective properties on gerbil models of bilateral carotid occlusion at doses of 50, 100, and 200 mg/kg [278].

Diabetes mellitus is a chronic metabolic disorder estimated as the fifth leading cause of death globally. Unfortunately, many anti-diabetic drugs show limited activity and the need for a new therapeutic strategy is increasing. Seaweeds possess interesting bioactive natural products with potential anti-diabetic effects such as polyphenols, sulfated polysaccharides, bromophenols, fucosterol, and carotenoids. These bioactive products may prevent the development of diabetes or modify the metabolic abnormalities through a large number of mechanisms [305]. Phlorotannins from *Ecklonia kurome*, polyphenols from *Ascophyllum nodosum*, and butyl-isobutyl-phthalate extracted from *Saccharina japonica* decreased postprandial hyperglycemia in vivo [285,286,287]. Similar results were reported for *Sargassum coreanum* extract and dieckol isolated from *Ecklonia cava* with an additional improvement of insulin sensitivity in streptozotocin-induced diabetic mice [288,289]. A diet rich in fucoxanthin, which is present in edible brown seaweeds such as *Eisenia bicyclis* and *Undaria pinnatifida*, could significantly reverse the white adipose tissue weight gain in fat mice and ameliorate blood glucose levels [290]. Similar results were reported for sulfated polysaccharides present in red seaweeds *Gracilaria birdiae* and *Plocamium telfairiae*, mainly due to their ability to decrease the rate of differentiation of 3T3-L1 cells into adipocytes [291,292]. Supplementation by the aqueous extract of the green seaweed *Ulva lactuca* rich in sulfated polysaccharide was effective in attenuating lipid peroxidation process and therefore decreasing TG, LDL, and VLDL and increasing HDL [293]. In contrast, Oh et al. [306] reported that brown seaweed consumption did not prevent long-term high fat diet-induced obesity in mice, but in turn was able to reduce insulin resistance and decrease the release of pro-inflammatory cytokines. Caulerpenyne extracted from *Caulerpa taxifolia* competitively inhibits pancreatic lipase activities and in vivo oral administration demonstrated a reduced and delayed peak plasma triacylglycerol concentration [307]. Park et al. proved that fucoidan stimulates lipolysis by inducing hormone-sensitive lipase expression and therefore has a high lipid inhibition activity [294]. Fucoxanthin affects lipid metabolism by manipulating the gene expression of enzymes involved in hepatic lipogenesis [295]. *Saccharina japonica* was found to possess some prebiotic effects and therefore could make desirable changes in intestinal microbiota composition that were significantly linked to a decrease in TG [296].

In vivo anticoagulant effects were also revealed for marine carbohydrates such as S-galactofucan from the brown seaweed *Spatoglossum schroederi* and spirulan from the blue-green algae *Arthospira pratensis* by increasing the clot formation time [297,298,299]. Sulfated polysaccharides isolated from *Monostroma angicava* effectively prolonged clotting time due to their high fibrinogenolytic and thrombolytic properties [300]. It should be noted that the in vivo antibacterial effects of seaweeds is often only achieved at toxic concentrations [5].

### 4.3. Seaweeds: Clinical Evidence

The results of the effect of seaweed on lipid profile supplementation in patients with type 2 diabetes indicated that seaweeds may be effective in lowering TG and LDL with a significant increase in HDL [308].

A prospective, open label clinical study was conducted to evaluate the oral administration efficacy of fucoidan in twenty advanced cancer patients with metastases. Results showed that fucoidan decreased levels of IL1β, IL6, and TNFα after two weeks of ingestion. Therefore, fucoidan could be act as supportive care for cancer patients undergoing chemotherapy [309]. Similar results were found during the investigation of a relationship between low incidences of breast cancer in Japan with the daily consumption of seaweed, where fifteen healthy postmenopausal women were recruited in a single-blinded placebo-controlled study using capsules containing *Undaria pinnatifida* for three months. At the end of the study, lower levels of urokinase-type plasminogen activator decreased significantly, and probably reduced cancer cell proliferation and invasiveness [310].

Alginate implants have received official FDA qualification for use in patients with an enlarged left ventricle. These implants were found to help in reducing the wall stress and relieving the muscle tension, thus preventing further dilation. The results of this study still need to be confirmed in larger clinical trials [311]. Additionally, alginate-hydrogel injections in combination with standard therapy were found to be more effective in improving symptoms and exercise capacity in patients with advanced chronic heart failure [312].

## 5. Seaweeds Quality Parameters and Health Safety

### 5.1. A Brief Overview of Regulatory Practices

Seaweeds are known as a rich source of essential elements since they have the ability to concentrate them from seawater. Aside from micro- and macroelements, marine macroalgae have an ingrained tendency to absorb and accumulate heavy metals from water over time (e.g., cadmium (Cd), coper (Cu), manganese (Mn), niquel (Ni), lead (Pb), zinc (Zn), mercury (Hg)) [313]. Although some of these metals like Zn and iron are essential for the growth and metabolic processes of seaweeds, excessive concentrations can result in toxicity and have a negative impact on growth, a reduction of the photosynthetic activity (leading to an abrupt decrease in pigment), and ultra-structural modifications [314]. Special attention should be paid to heavy metals, iodine, radioactive isotopes, ammonium, and organic compounds such as pesticides and dioxins [34,315]. The excessive consumption of seaweeds, especially those that contain elevated levels of non-essential metals, may constitute a health risk [316]. Therefore, the application of seaweeds as nutraceuticals, cosmetics, foodstuff, etc. should be preceded by the detailed chemical analysis including the content of toxic metals such as arsenic, cadmium, chromium, lead, mercury, nickel, zinc, etc., in order to assess the safety of this raw material [52,314,317].

Seaweed species and their components used as food must meet certain consumer safety regulations [34] in terms of toxicological and bacteriological criteria [318]. For example, for seaweed species (brown seaweeds *Ascophyllum nodosum, Laminaria digitata, Saccharina latissimi, Himanthalia elongata, Undaria pinnatifida, Fucus vesiculosis*; green seaweeds *Ulva* spp.; and red seaweeds *Palmaria palmata, Porphyra umbilicalis, Pyropia tenera, P. yezoensis, P. leucosticta, Porphyra dioica, P. purpurea, P. laciniata, Gracilaria gracilis,* and *Phymatolithon calcareum* [318]) that are authorized in France (the first country in Europe) for direct consumption as sea vegetables or food ingredients, the following quality criteria are required: (1) Toxic minerals such as arsenic (inorganic), cadmium, lead, mercury, iodine, tin; and microorganisms (aerobes, anaerobes, fecal coliforms, *Clostridium perfringens*) [316]. It should be underlined that the content of contaminants is seaweeds is regulated in various ways by different legislation globally.

### 5.2. Multidimensional Procedures to Overcome Seaweeds’ Antinutrients

Considering human health, seaweeds should be also examined in terms of the content of antinutritional factors. Antinutrients can contribute to a reduction in the nutritional quality of seaweeds because they can reduce the digestibility and bioavailability of health-advancing ingredients [319,320]. Seaweeds contain antinutrients such as phytic acid, polyphenol compounds (tannins), lectins, amylase, and trypsin inhibitors [321].

Lectins are regarded as antinutritive and/or toxic substances that may survive digestion by the gastrointestinal tract of consumers. As they bind to membrane glycosyl groups of the cells lining the digestive tract, they trigger harmful local and systemic reactions [322]. In order to assess the toxic potential of lectins, their reactivity with human and animal erythrocytes should be examined [321]. In the work of Oliveira et al. [321], it was shown that the mixture of washed-up seaweeds (consisting of 24 red, nine green, and four species of brown algae) contained low levels of lectins: 32 and 64 HU (the reciprocal of the highest dilution exhibiting hemagglutination) per gram of meal for chicken and rabbit trypsin-treated erythrocytes, respectively. Besides lectins, tannins, polyphenols, or lipids (present mainly in brown seaweeds) may also interfere in agglutination processes as they can strongly agglutinate erythrocytes [321,322].

Other antinutritional compounds are phytic acid, tannins, trypsin, and alpha-amylase inhibitors that can negatively influence the digestibility and/or bioavailability of some nutrients such as proteins and trace minerals [321,323]. Oliveira et al. [321] showed that the mixture of 37 species of washed-up seaweeds dominated by red seaweeds (24 species) collected from the coast of Ceará (Brazil) contained low levels of phytic acid (0.45%), higher tannins (59 mg/100 g), and a high level of trypsin (99.0% inhibition) and alpha-amylase inhibitors (70.5%). Reduction of the level of antinutrients (e.g., tannins, phytic acid) and an improvement in seaweed protein and starch digestibility can be achieved by thermal heat processing of the biomass (e.g., cooking at 121 °C for 10 min) [323].

Seaweed polysaccharides are also regarded as an antinutritional factor that limits the action of human digestive enzymes, for example, pepsin activity. The presence of anionic polysaccharides (e.g., alginates or carrageenans) and neutral polysaccharides (e.g., xylanes and cellulose) in large quantities in seaweed cell walls can limit the availability of biologically active compounds (e.g., proteins) not only during enzyme digestion, but also during extraction procedures that aim at the isolation of bioactive compounds from algal biomass [324]. Therefore, the presence of a large polysaccharide fraction (e.g., cellulose) in seaweeds can limit their use as a source of proteins for humans or animals [318]. One of the procedures that aim to overcome the antinutrients in seaweeds is biomass pre-treatment in which cells are disrupted, making biologically active compounds more bioavailable. Methods of seaweed cell wall disruption include mechanical, physical, thermal, chemical, and enzymatic algal biomass treatments (e.g., bead mills, freezing and thawing, osmotic shocks, homogenization at high pressure, ultrasonication, microwave, autoclaving, alkaline lysis, sulfuric acid treatment) [325].

There are some methods that allow for the removal of antinutritional factors from seaweeds, for example, “enzymatic liquefaction”. This is applied to especially fresh (not dry) seaweeds to eliminate xylan and cellulose, which will improve algal protein digestibility [326]. For this reason, several enzyme combinations are used, for example, purified enzymes such as endo-β-1,4-d-xylanase or commercial preparations such as “Celluclast”. In this process, the seaweed texture is modified in order to produce an intermediate food that is usable for further industrial transformation. The main disadvantage of this enzymatic process is the availability of enzymes necessary for the degradation of the polysaccharides [324].

Another option to reduce the content of antinutritional factors, increase the digestibility of seaweeds, and at the same time improve the nutritional value of algal products is fermentation with the use of lactic bacteria, yeast, or other fungi [318,325,327]. Ardiansyah et al. [327] showed that there is a possibility of reducing the content of anti-nutritional factors when the fermentation process is used. In the case of the fermentation of brown seaweed *Sargassum* sp. with *Lactobacillus* spp., *Aspergillus niger,* and *Saccharomyces cerevisiae*, the reduction of phytic acid, tannins, saponins, and total polyphenols was observed. The best results in terms of carbohydrate content reduction in the fermented *Sargassum* sp. were observed for the use of *A. niger*, phytic acid for *A. niger*, total polyphenols for *Lactobacillus* spp., tannins for *S. cerevisiae* and *Lactobacillus* spp., and saponins for *Lactobacillus* spp. Reduction of the content of phytic acid in seaweeds (e.g., through the fermentation process) can be beneficial because phytate can inhibit digestibility by chelating with calcium binding to the substrate or proteolytic enzyme [327]. The reduction of phytic acid and tannin may result from the production of the enzyme alpha-galactosidase by microorganisms that are used as a starter culture for fermentation [327].

### 5.3. Bioavailability of Seaweed Bioactive Compounds

Seaweeds serve as a rich source of biologically active compounds. Among them we can distinguish nutritional elements, polysaccharides (e.g., alginates, fucoidan, laminarin, mannitol, phorphyran, carrageenans, agar, ulvan), proteins, peptides and amino acids, lipids, fatty acids, phospholipids, glycolipids, pigments (chlorophylls, carotenoids, phycobiliproteins), phenols, and other phlorotannins [34,325,328]. Although algae are a source of these compounds, it does not mean that they are bioavailable to humans. Bioavailability is an important factor in nutrition that varies with gastrointestinal conditions [317]. The bioavailability of seaweed bioactive compounds mainly depends on their complex polysaccharide structure, which may impede the accessibility of compounds to the gastrointestinal enzymes as well as on the content of antinutritional factors described above [317,318,324,329]. Digestion, absorption, transport, utilization, and elimination are other processes that influence the bioavailability of algal compounds [317]. Therefore, there is a necessity to conduct more research regarding the bioavailability of algal active compounds to humans. These experiments will allow the assessment of the safety of seaweeds and to gain knowledge on the real nutritional value of this type of foodstuff. Domínguez-González et al. [317] and Maehre et al. [329] proposed a good and inexpensive in vitro digestion procedure (simulated gastric and intestinal digestion/dialysis) to assess the bioavailability of compounds. In the case of mineral bioavailability, the form in which they exist and their solubility can be studied [330].

The main factor that limits the bioavailability of active compounds from seaweeds is polysaccharides in the cell wall. Very often, they hinder the isolation of bioactive metabolites. In order to facilitate their release, the biomass can be pre-treated using mechanical, physical, thermal, chemical, and enzymatic methods. This procedure is very often used before the extraction of compounds from algae [325]. Obtained seaweed extracts can be used in phytopharmacology and cosmetic formulations [245,325,331].

Another important issue that should be considered is the digestion of seaweed polysaccharides. Human intestinal enzymes cannot entirely digest them. Therefore, it is very often considered as a source of dietary fiber [22,25]. Fiber components such as cellulose, hemicellulose, other polysaccharides, and lignin, can form insoluble complexes with minerals, and as a consequence, reduce their bioavailability [316,330,332]. For example, there is a strong affinity of divalent cations (particularly calcium) for polysaccharides with carboxylic groups (e.g., alginates) that can limit their bioavailability [316]. On the other hand, seaweed polysaccharides are able to retain cholesterol and related active compounds and inhibit their absorption in the gut [332]. Moreover, seaweeds, due to the possibility of binding metal ions from the marine environment, can serve as a bioavailable source of micro – and especially macroelements in digestion processes [313].

In the literature, there are some examples concerning seaweed biomass as good sources of bioavailable compounds such as ascorbic acid and iron [333], cobalt, chromium, manganese and vanadium [317], protein [329], protein, macro-elements Mg, Ca, Na, K and microelements Fe, Zn, Cu [327], and minerals [330]. Most of them concern the bioavailability of elements, because they play an important role in many biochemical reactions in the body, mainly as cofactors of enzymes [330]. García-Casal et al. [333] examined the bioavailability of iron in humans from different species of marine algae: green *Ulva* sp., brown *Sargassum* sp., and red *Porphyra* sp. and *Gracilariopsis* sp. Besides iron, which is essential to combat anemia, vitamin C and phytic acid content were also examined in the algal biomass. Phytates, which are known as an antinutritional factor, were not detected in all seaweeds. Ascorbic acid improved iron absorption from seaweeds. In this study, it was shown that the addition of algae (raw or cooked) to the rice-based meals of subjects, increased iron absorption and cooking did not affect the iron when compared with raw seaweeds. However, absorption was higher for cooked than for raw algae.

The possible effects of seaweed processing on the bioavailability of bioactive components is largely unknown; the literature contains few examples. Heat treatment is one of the methods that increases the bioavailability of certain nutrients and inhibits anti-nutrients of seaweed origin. However, some compounds that are sensitive to high temperatures, for example, free amino acids or vitamins, can be lost [329]. Heat treatment can also induce protein denaturation, which can make minerals insoluble and therefore not bioavailable [330].

Maehre et al. [329] examined the effect of heat treatment on the bioaccessibility of proteins from brown *Alaria esculenta* and red *Palmaria palmata* seaweed. An in vitro gastrointestinal digestion model showed that a short heat treatment (boiling) of *Palmaria palmata* increased the amount of bioaccessible protein, with no deterioration of the amino acid composition. Preparation of food containing seaweeds and its cooking can influence the final content of some elements (e.g., iodine) in the product. Reduction of iodine content in seaweeds up to 70% after five minutes of boiling in water was noted by Duinker et al. [315]. A rapid release of iodine from seaweed can result from the weakness of the linkages between polysaccharides and iodine [334]. It is worth mentioning that iodine lost from seaweeds during boiling can be reintroduced, the boiling water can be consumed by inclusion, for example, in soup [315].

Since in many countries, the most common method of cooking seaweed is boiling, Santoso et al. [330] examined the effect of boiling Indonesian seaweeds in different solutions (e.g., water, acetic acid, and sodium chloride since the last two are very often used as seasonings) on the solubility of minerals such as Mg, Ca, K, Na, Cu, Zn, and Fe. Boiling of several seaweeds, green *Caulerpa racemosa* and *Ulva reticulata*, brown *Padina australis, Sargassum polycystum* and *Turbinaria conoides* and red *Kappaphycus alvarezii*, increased the solubility of examined minerals, especially that of Ca.

Another simple and cheap method that can enhance the nutritive value of seaweeds and at the same time improve their digestibility is fermentation [325,327,335]. Uchida and Murata [335] found that the fermentation of algal biomass with lactic acid bacteria and yeast could considerably decrease the content of crude fiber, increase protein value, enrich with essential amino acids, essential fatty acids, minerals, and vitamins as well as improve the digestibility of seaweed-based food. Ardiansyah et al. [327] also showed that there is a possibility of improving the nutritive quality of seaweed when the fermentation process is used. In the case of the fermentation of brown seaweed *Sargassum* sp. with *Lactobacillus* spp., *Aspergillus niger*, and *Saccharomyces cerevisiae*, the level of crude proteins significantly increased, whereas the content of crude lipids and carbohydrates significantly decreased (probably due to their utilization by microorganisms). Fermented seaweed was also a source of bioavailable macro- (Mg, Ca, Na, K) and microelements (Fe, Zn, Cu). The best results in terms of the content of crude proteins and macroelements Mg, Ca (increase), and carbohydrate and phytic acid (decrease) was observed for the fermentation of *Sargassum* sp. with *A. niger*. Further evaluation of the bioavailability of active compounds from seaweeds is still required.

## 6. Targeting Seaweed Potentialities for Cosmetic Purposes

The most ancient record of agarose extraction dates back to the mid-17th century [65]. The algal industry started in China in the 1950s. Macroalgae were considered as a topical application against infections, especially from mycobacterium against scrofula; in tracing back the origin of this use, Ancient Greek documents seem to be present [336].

Recent advances in marine biotechnology offer great help in studies on ageing, inflammation, and skin degradation. At the same time, dermatological research suggests that marine bioactive ingredients may have great benefits for the treatment of skin disorders. In a recent interesting review, a complete tabulated list of more than one hundred algae species and compounds extracted with cosmetic properties and related products, is reported [65]. The topic is still in full expansion, a simple search in the CORDIS database [337] reveals a list of dedicated projects in Europe, specifically dedicated to this field (TASCMAR, ALGAECOM, etc.). Indeed, regulated substances and ingredients in the EU market can be found in the CosIng database, which contains all data since the adoption of the Cosmetics Directive in 1976 [338].

Cosmeceuticals are defined as preparations containing various bioactive ingredients, thus it seems that the best way to describe the potentialities of seaweeds for cosmetic purposes is to start listing chemical compounds of interest to this industrial sector. However, whole macroalgae extracts are in use for nutritional, nutraceutical, and cosmeceutical benefits for the various active ingredients that are present in the extracts from algae. Indeed, extracts of *Saccharina latissima*, a species of brown algae, are sold under the tradename Phlorogine by Biotech Marine, France [339].

The importance of seaweed in the cosmetics industry is witnessed by the estimation that it makes up almost 40% of the world’s hydrocolloid market, as reported by studying the size of the international market for seaweed and its commercially important extracts [340]. Among the inorganic compounds, sea salts are recognized agents for a series of effects in cosmetics [341]. A list of biomolecules featuring a carbon skeleton (organic compounds) follows to describe the macroalgal molecules of interest for the cosmetics industry.

### 6.1. Enzymes, Collagen, and Bioactive Peptides

The protein class of compounds is well represented; for enzymes, a number of experimental results including those from studies on humans have indicated their role as antioxidants and the delicate balance of these molecules that is required at the skin level. A practical approach for the use of these molecules is by supplying them through topical application (sun products, sunscreen, etc.), although stabilization of these ingredients could be and actually is, an issue. One of the proposed solutions in this context is the use of anti-oxidant enzymes originating from organisms isolated from extreme conditions, the “extremophiles”. Marine examples of these organisms are well studied. Phycoerythrin, a protein acting as a photosynthetic accessory pigment in red algae, has biotechnological applications as a natural colorant suitable for formulations in cosmetic products. The suitability of its production from culture is of great interest [342]. Collagen represents another example of this class of biomolecules. The field of manufacturing this molecule is the need for alternatives to animal sources, and marine resources such as scales, skin, and bone that are readily available in seafood waste represent useful originating material for collagen hydrolysates. This component has been studied for effects related to biophysical skin parameters, moisture, and sebum as bioactives in cosmetics [343]. Mycosporine-like amino acids are water-soluble compounds possessing a photo-protective action. They also have antioxidant and skin protective action and the interest for possible medicinal and cosmetic applications is clearly based on this aspect. A new method for quantitative analysis in marine species has been developed [344] and recent patents on this subject have been reviewed [345]. Lectins, carbohydrate-binding proteins, have been found in a wide range of organisms, and algal lectins could have anti-inflammatory [346] or antioxidant activity [347]. This easily accessible marine source is also attractive as a protein source for the industrial production of functional peptides in the cosmetic domain.

### 6.2. Fatty Acids and Derivatives

Total lipid content of seaweed is thought to be smaller when compared to microalgae or terrestrial plants, however, specific bioactivities featuring lipid components and the biodiversity of wild seaweeds and possible aquaculture facilities also make these algal components of interest from a biotechnological point of view. The main interest in cosmetics is based on the free radical scavenging capability of lipid molecules. Basic research demonstrates that all fractions of lipids exhibit this activity, the most being associated to neutral fractions, and a biorefinery approach valorizing these seaweed components is an interesting point [121]. Green algae are similar to plants in fatty acid composition with a higher oleic and alpha-linoleic acid content. Red algae have a high eicosapentaenoic acid content, a substance mostly found in animals, especially fish. In an interesting report on temperature requirements for the growth and survival of macroalgae from Disko Island (Greenland), the authors pointed out the specific composition of polyunsaturated fatty acids [348]. Glyco- and phospholipids have applications in cosmetic and beauty products as studied for microalgae originating components of this type [349]; similar compounds from macroalgae could be advantageous and should not be neglected.

### 6.3. Phlorotannins and Polyphenols

Polyphenols are among the most abundant antioxidants in the human diet. They derive from a long list of food types (fruits, vegetables, cereals, olive oil, dried vegetables, chocolate), and beverages (coffee, tea, and wine). As cosmeceutical components, an increasing number of studies has evaluated their antioxidant effects while as nutricosmetics, a reduction of UV-induced damage and hyperpigmentation in women seem activities of interest. The presence of phenolic compounds in marine plants, namely in algae [350] has been well studied. Brown seaweed accumulates a variety of phloroglucinol-based polyphenols (phlorotannins), *Ecklonia cava* is a particularly rich source of phlorotannins compared to other brown algae, bioactivities span from antioxidant, antibacterial, anti-inflammatory, and anti-allergic. Regarding phloroglucinol, it is well known in cosmetics, textiles, pesticides, paints, and cement [351]. Phlorotannins have antiallergic, bleaching, anti-wrinkle, and skin antiaging activities, of interest in the cosmetic industry. More than 100 metabolites, belonging to different classes are known, being present in 49 species of brown algae [65].

### 6.4. Polysaccharides

Many types of biologically active polysaccharides are present in seaweeds. Generally, specific chemical properties of these compounds are exploited when they are used as ingredients in cosmeceuticals, with moisturizing and antioxidant capacities at the top of this list. Gelling property and action as stabilizers in emulsions are also usually acknowledged, but recently, specialized applications in drug release, in vaccine production, as anti-thrombogenic, anticancer, and antiadhesive agents have also been reported in the literature [352]. Agar is used in creams and as an emulsifier and stabilizer in many cosmetic products. Alginates possess chelating properties and are widely used as thickeners or gelling agents in drugs and cosmetics, as emulsion stabilizers in hand jelly and lotion preparations, as ointment bases in greaseless creams, dentifrices, and other products. An interesting application is in the formulation of skin protective beauty masks for which alginates produce a flexible film with improved adhesion to the skin. Carrageenans are used in toothpaste, lotions, hair conditioners, lotions, medicines, sunray filterers, shaving creams, shampoos, deodorants stick, sprays, and foams. The sulfate polysaccharide porphyran is useful as a skin whitening agent for the cosmetics industry while laminaran is signaled in anticellulite cosmetics products. Fucoidan action as a preventive for photoaging is based on the inhibition of metalloproteinase, which is induced by UVB radiation, but this polysaccharide is useful as an inhibitor of tyrosinase, reducing skin pigmentation and used in skin whitening agents [353]. Among the other effects, the elimination of free radicals, reduction of inflammation, wrinkles, allergies, and sensitive skin reaction, have also been reported. Interestingly, fucoidan is also able to increase fibroblast proliferation and collagen and the deposition of other matrix factors [354]. Ulvan features chemical and physicochemical properties that make this class of polysaccharide useful agents for cosmeceuticals and in other fields [78]. The regular structure containing uronic acids, sulfate groups, and rare sugars (rhamnose, iduronic acid) is related to its functional properties. New mechanisms controlling the atypical gelling could certainly be of interest. Although chitosan is a known crustacean originating polymer, a recent report depicted some origin from seaweed [355] and known Chinese originating products are registered as “chitosan from seaweed extract”. Different functional roles as a skin care and hair care ingredient or as an oral hygiene agent and as a carrier for active compounds are ascribed to chitin and chitosan. The importance of the physico-chemical properties of the polymer are clear for their use in cosmetics [356], most of the functions of interest for cosmetics are reported in the EU database CosIng (see above). Besides chitin or chitosan, several derivatives of both polymers have been synthesized and tested against different bacterial strains. Furthermore, polysaccharides of interest for the cosmetics industry could be elicited by low molecular weight oligosaccharides from marine origin, an example from the red algae *Pyropia yezoensis* appeared recently in the literature [357] with emphasis on interesting bioactivities such as antioxidant, antitumor, antifatigue, anti-inflammatory, and protection against UVA-induced photoaging.

### 6.5. Carotenoid Pigments, Fucosterols, Tocopherols, and Terpenoids

Carotenoids are useful as a source of pigmentation; moreover, their function as an antioxidant and a role in cancer prevention add beneficial effects to human health; in the skin, they have an important role in photoprotection against UV radiation. Indeed, these compounds are used in pharmaceuticals and nutraceuticals. According to the results of studies of these molecules applied in cosmetics, a significant enhancement in the elasticity of skin and a more pronounced cutaneous hydration were observed [358]. Fucosterol, as a terpenoid extracted from *Ecklonia stolonifera*, shows among others, strong antioxidant activity [359]. Fucoxanthin may be an effective ultraviolet protector able to be used in cosmetics and sunscreen to prevent photoaging [360]. The tocopherol content of seaweed has been studied [361], but microalgae seem to produce an important amount of this component. It is effective for skin protection and is believed to play a major role in the prevention of light-induced pathologies of skin and eyes [362]. Tocopherol is used in many industrial applications, as a food preservative, sunscreens in cosmetics, etc. [363]. A family of diterpene glycosides known as pseudopterosins that are isolated from the Caribbean gorgonian coral *Pseudopterogorgia elisabethae* is effective in preventing sun damage to the skin and nourishing the skin, and is currently incorporated into skin care preparations [342,364].

## 7. Seaweeds as Promising Cosmetics: What will Happen?

In an old book [365], a chapter was dedicated to the exploitation of natural product diversity including fragrances and cosmetics of marine originating products. This field has since been increasingly developed; however, the foundation of consumers’ expectations is still based on the “natural” quality that is perceived as safe, regardless of the terrestrial or marine origin of the product, even though some clear differences in consumer expectation could be envisaged across the markets of different countries. However, it seems clear that at least the three keywords of wellness, clean ingredients, and natural quality of products could label global expectations. New discoveries regarding new products have elicited new interest from the consumer side, acknowledging specific concerns (e.g., solutions for photoaging). Indeed, research studies aimed at understanding photo-acclimation mechanisms in different marine flora and fauna can deliver potent compounds for their beneficial role in cosmeceutical and pharmaceutical applications [366].

Recent advances in blue biotechnology research in terms of science, technology, innovation, and policy for the development of the field have been well addressed [367] and how marine biotechnology may provide new solutions to some of the grand challenges faced by our society has been well considered. New successes will depend on a number of new bioproducts discovered, on the possible sustainable use, and on the optimization of production, topics that can efficiently be considered with the integration and collaboration of multidisciplinary teams of scientists. In a paragraph discussing resource assessment and harvesting strategy, Dhargalkar and Verlecar pointed to these problems [368] for seaweed, especially from the Southern Ocean. Their conclusion can be agreed upon, if to assess the economic feasibility of this marine resource as promising cosmetics, “more detailed statistically valid system of surveys, eco-biological studies, proper biomass estimation, standing stock, and seasonality of harvesting” have to be considered.

Under a different point of view to the question “what will happen?”, a reasonable answer should point to aging as an international social, economic, and individual topic of importance in global science. Its prevention and decreasing of side effects are certainly future challenges; in particular, the role of microbiota at the skin and general levels is of great interest. Additional points to be included here are related to the safety of products, consumer protection, and different market regulations. All three will certainly be developed along the already present auspices with major similarities among the regulations present in different countries with the aim of better protection for the typical worldwide consumer who should be informed with better scientifically sound material.

## Figures and Tables

**Figure 1 molecules-24-04182-f001:**
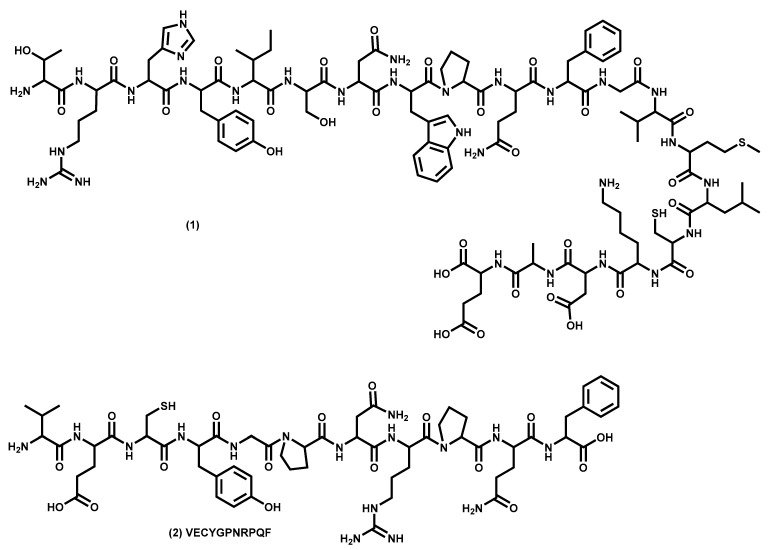
Primary structure of a hypothetical protein consisting of all amino acids (**1**) and the seaweed peptide with potent antioxidant activity (**2**).

**Figure 2 molecules-24-04182-f002:**
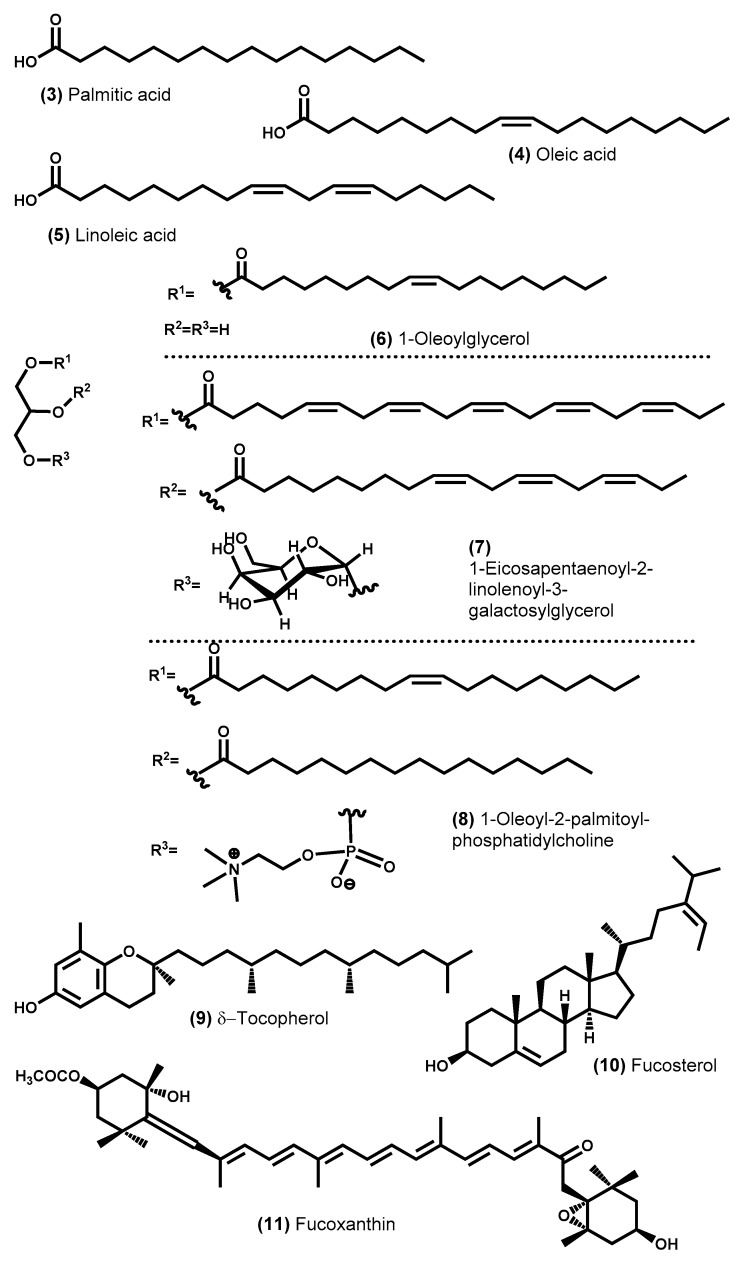
Chemical structure of some examples of saturated and unsaturated fatty acids (**3**–**5**), acylglycerols (**6**), glycoglycerolipid (**7**), phospholipids (**8**), lipophilic vitamin (**9**), sterol (**10**), and carotenoid (**11**) identified in seaweeds [48,119,120].

**Figure 3 molecules-24-04182-f003:**
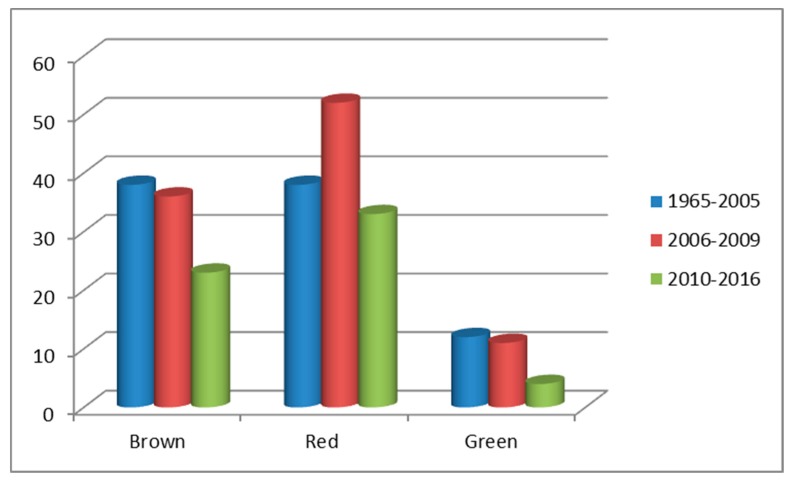
Distribution of the metabolites isolated from macroalgae per year, adapted from Blunt et al. (2009) and completed with data from the same authors [138,139,142,143,144,145,146,147,148].

**Figure 4 molecules-24-04182-f004:**
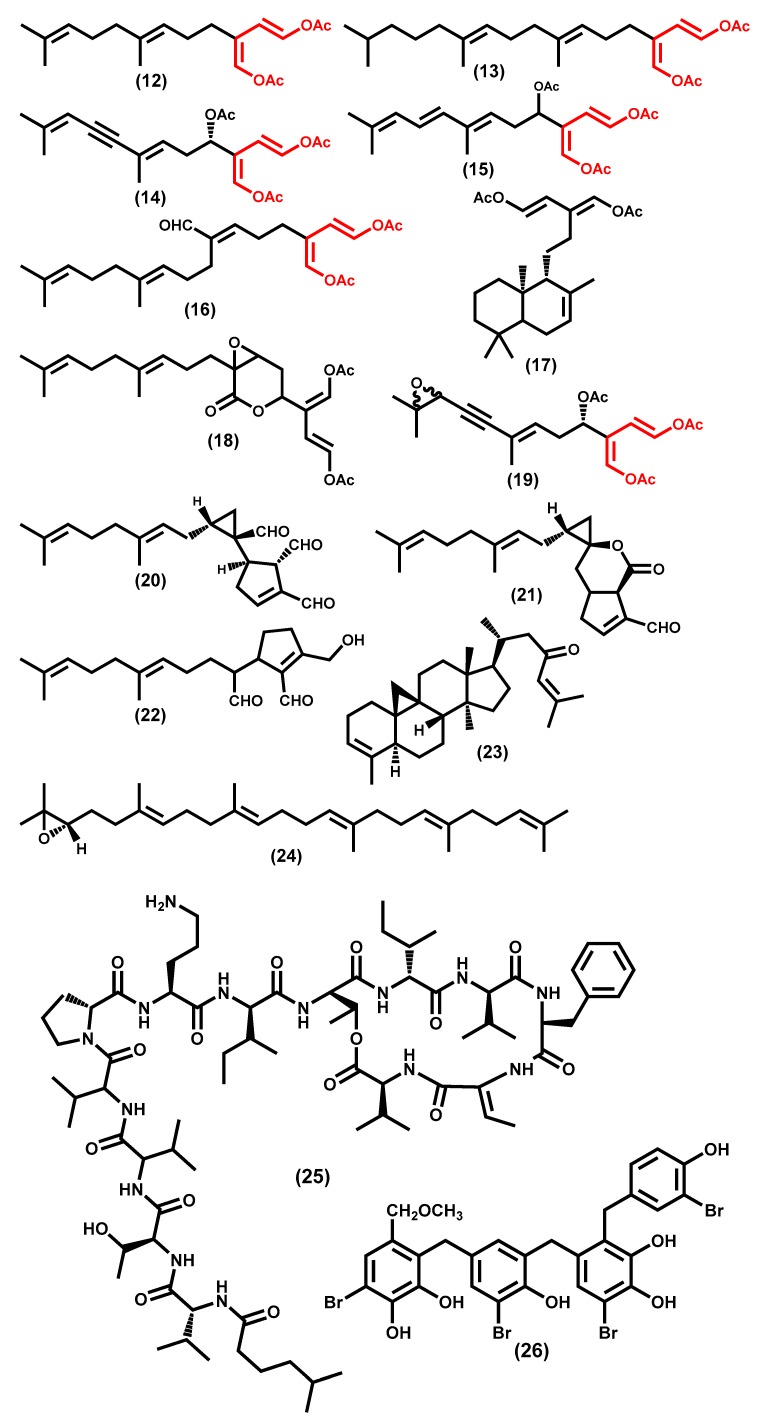
Chemical structure of secondary metabolites from green algae. (Ac=CH_3_CO).

**Figure 5 molecules-24-04182-f005:**
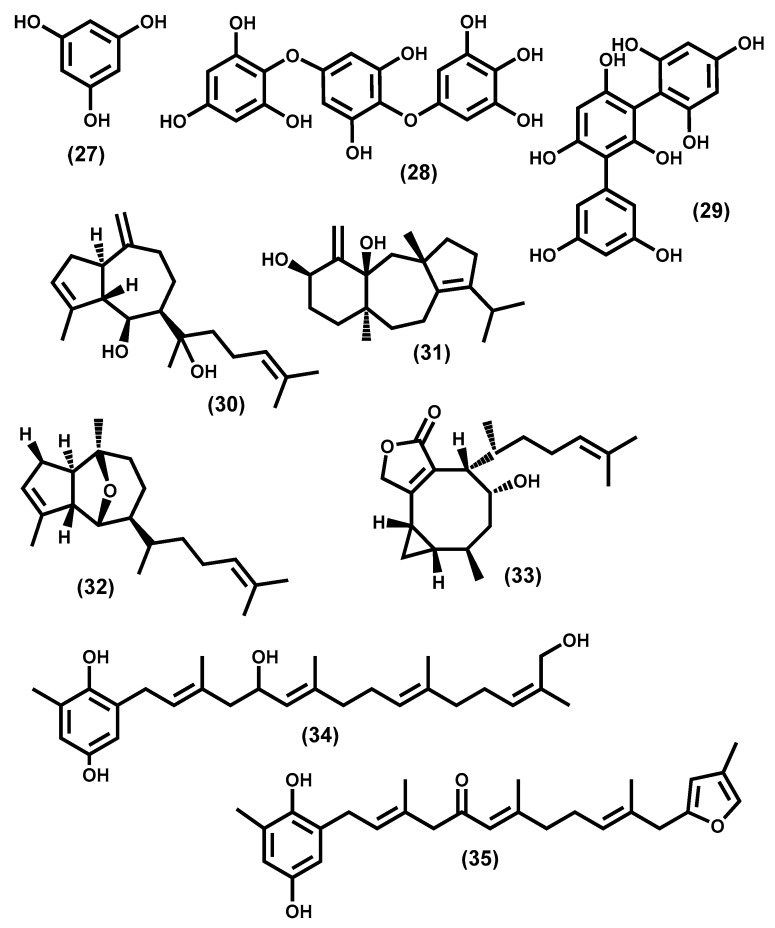
Chemical structure of secondary metabolites from brown algae.

**Figure 6 molecules-24-04182-f006:**
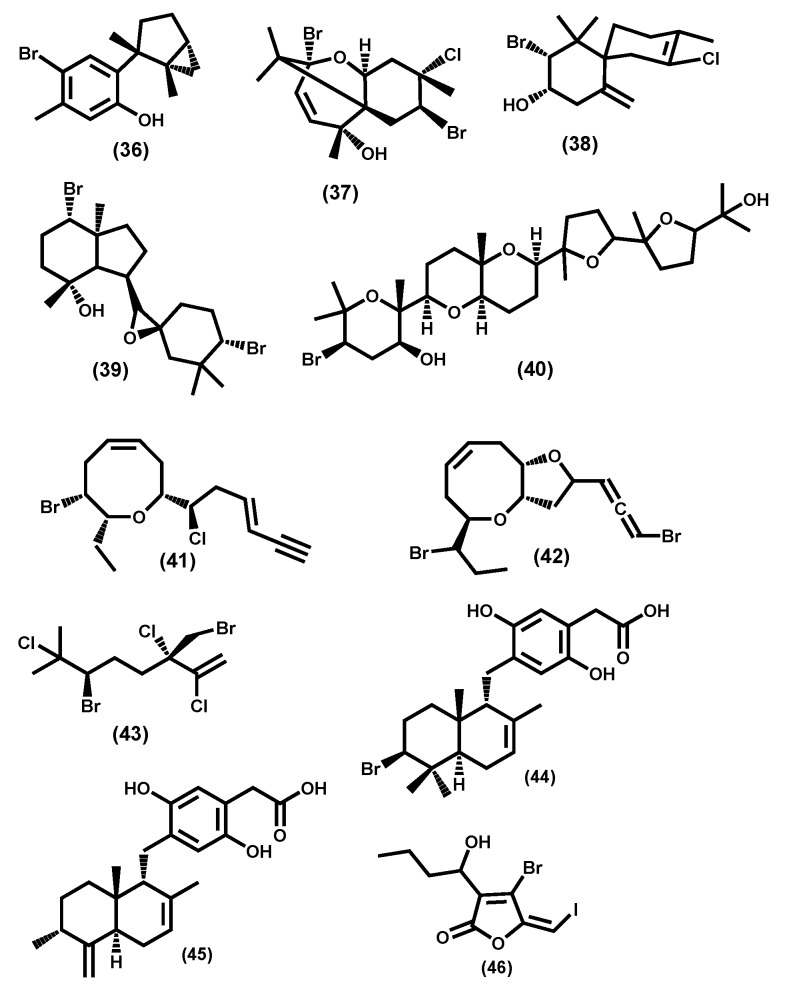
Chemical structure of secondary metabolites from red algae.

**Figure 7 molecules-24-04182-f007:**
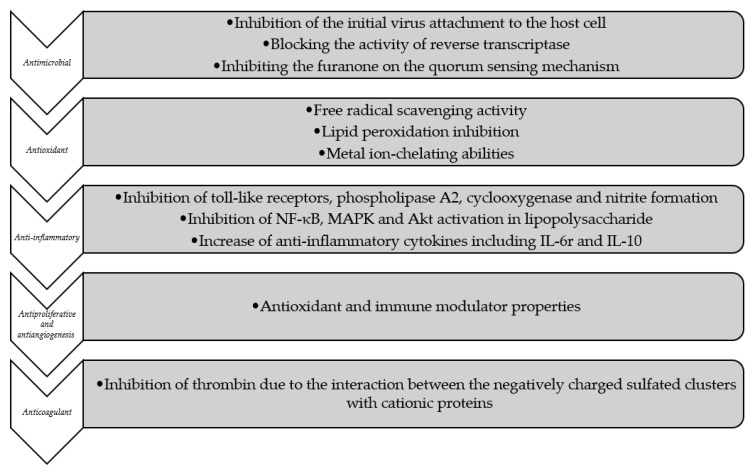
In vitro mechanism of the action of bioactive compounds isolated from seaweeds.

**Table 1 molecules-24-04182-t001:** Average content of each seaweed primary metabolite class and minerals.

Metabolites Classes	Green (Chlorophyta)	Brown (Ochrophyta)	Red (Rhodophyta)
Total Carbohydrates (% dw)	29.8–58.1 [29,31,32]	12.2–56.4 [31,32,33]	34.6–71.2 [29,31,32]
Minerals (ash,% dw)	11–73% [29,31,34,35,36]	17–44% [31,34,36,37,38,39]	7–37% [31,34,36,37,39]
Macrominerals (% dw)	0.2–2.4	1.3–7.0	0.3–10
Na	1.1–2.8	0.9–11.5	0.2–9.2
K	0.09–4.9	0.4–3.0	0.2–1.9
Ca	0.4–3.7	0.1–1.2	0.1–1.7
Mg	0.09–0.25	0.07–0.4	0.1–0.7
P	[31,35,36,40]	[31,36,39,40]	[31,36,39,40]
Trace elements (mg/kg dw)	98–5800	16–1854	366–2110
Fe	3.0–32.7	6.8–154	23–72
Zn	0.5–32.7	0.8–38.6	1.99–34.6
Cu	26–180	2.7–547	4.1–392
Mn	23–480	59–11096	22–340
I	[31,35,36,40]	[31,36,38,40]	[31,36,40]
Proteins (% dw)	8.7–32.7 [29,31,41,42,43]	4.3–24.0 [28,30,31,38,41,42,43,44]	8.0–47.0 [28,29,31,41,42,43,44]
Lipids (total lipids%, dw)	0.2–4.1 [16,29,31,34,44,45]	0.3–4.5 [16,30,31,34,37]	0.12–3.8 [16,29,31,34,37,44,45]
Saturated fatty acids (% of total fatty acid)	34–77 [28,34,46]	20–50 [31,34]	25.5–76 [29,34,47]
Mono-unsaturated fatty acids (% of total fatty acid)	12.2–23 [29,46]	11–36 [30,34]	10.6–35.7 [29,34,47]
Poly-unsaturated fatty acids (% of total fatty acid)	6.6–39 [29,34,46]	20–67 [30,34]	12.1–68 [29,34,47]

% dw, g/100 g on a dry weight basis; nd, not determined.

**Table 2 molecules-24-04182-t002:** Chemical composition and structural features of polysaccharides from seaweeds.

Polysaccharides, Distribution (Function)	Monomeric Composition * (Glycosidic Linkage)	Other Groups Substituents	Other Characteristics
Fucoidans Brown algae (structural)	Type I: L-Fucose (α-1→3) Type II: L-Fucose alternating (α-1→3 and α-1→4)	Sulfated at C-4 (mainly) and C-2 Sulfated at C-2 and at alternating units also at C-3	Molecular weight, degree of branching, sulfation ratio and presence of other monosaccharide (xylose, galactose, uronic acid) variable with the species
Agars Red algae (mainly structural)	Agarose type: β-d-Galactose, 3,6-anhydro-α-l-galactose (alternating β-1→4; α-1→3); Agaropectin: alternating d- and l-galactose (alternating β-1→4; α-1→3)	Sulfate, methoxy, and pyruvate substituents at C-2, C-6 and C-4 or C-3	The degree of substitution like sulfation is species specific such as the proportion of agarose and agaropectin.
Carrageenans Red algae (mainly structural)	β-d-Galactose and 3,6-anhydro-α-d-galactose (alternating β-1→4 and α-1→3)	Sulfate group: only at C-4 of β-d-galactose (κ-carrageenan); accumulate with C-2 of 3,6-anhydro-α-d-galactose (ι-carrageenan)	Carrageenans content and type is species specific and depends on ecophysiological factors. There are other less abundant and less commercially relevant types of carrageenans (e.g., θ-, μ-, and β-carrageenans)
Ulvans Green algae (mainly structural)	A3s type: α-l-Rhamnose and β-d-glucuronic acid (1→4) B3s type: α-l-Rhamnose and α-l-iduronic acid (1→4)	Sulfate groups mainly at C-3 but also at both C-2 and C-3 at rhamnose units	Other sequences were determinate involving xylose and xylose-2-sulfate instead uronic acids residues (U3s and U2′s3s types) and a sequence like A3s but where glucuronic acid occur as branches on C-2 of rhamnose (A2g3s type)
Alginates Brown algae (mainly structural)	β-d-Mannuronic acid and α-l-guluronic acid (1→4)	Each monomer has a *terminus* COOH group	The monomers can be organized into homopolymers or hetero-polymers. The α-1→4 linkages give a bent and rigid chain while the β-1→4 linkage gives a flexible and linear chain.
Laminarin or laminaran Brown algae (storage)	d-Glucose (β-1→3)		Some residues of mannitol and/or uronic acids and occasional β-1→6-linkages in branching points as interchain linkages

* Monomeric composition of the main chain.

**Table 3 molecules-24-04182-t003:** In vivo pharmacological effects for bioactive compounds found in seaweeds.

Pharmacological Effects	Type of Seaweed	Ref.
Anti-inflammatory	Methanolic extracts of the brown seaweeds *Padina tetrastomatica*, *Dictyota menstrualis*, *Sargassum wightii*, and *Spatoglossum schroederi*	[272,273]
	Red seaweeds *Gracilaria edulis* and *Dichotomaria obtusata*	[255,274]
	Green seaweed *Caulerpa racemosa*	[274]
Antioxidant	Green seaweed *Ulva lactuca*	[275]
	Brown alga *Ecklonia cava* and *Pelvetia siliquosa*	[276,277]
	Aqueous extract of the green seaweeds *Halimeda incrassata*	[278]
	Fucoidan from brown seaweeds	[279,280]
Antitumor	Brown seaweeds *Undaria pinnatifida* and *Fucus vesiculosus*	[281,282]
	Red seaweeds *Chondrus ocellatus* and *Gelidiella acerosa*	[278,283,284]
Hypoglycemic	Brown seaweeds *Ecklonia kurome*, *Ascophyllum nodosum*, *Eisenia bicyclis*, *Undaria pinnatifida*, *Ecklonia cava*, and *Saccharina japonica*	[285,286,287,288,289,290]
	Red seaweed *Sargassum coreanum*, *Gracilaria birdiae*, and *Plocamium telfairiae*	[289,291,292]
Hypolipidemic	Brown seaweeds such as *E. bicyclis* and *U. pinnatifida*	[290]
	Red seaweeds *Gracilaria birdiae* and *Plocamium telfairiae*	[291,292]
	Green seaweed *Ulva lactuca* and *Caulerpa taxifolia*	[293,294]
	Fucoxanthin and Fucoidan from brown seaweeds	[294,295,296]
Anticoagulant	Brown seaweed *Spatoglossum schroederi* Blue-green algae *Arthospira pratensis*	[297,298,299]
	Green seaweed *Monostroma angicava*	[300]

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
