# Peer review of "Current Trends on Seaweeds: Looking at Chemical Composition, Phytopharmacology, and Cosmetic Applications"

_molecules, 2019, doi:10.3390/molecules24224182_

Round 1

Reviewer 1 Report

The manuscript title is  Current trends on seaweeds: looking at chemical composition, phytopharmacology and cosmetic applications. The review from this point of view is rather complete. However, the author spent more time and word on chemical composition and  phytopharmacology. It is top-heavy. In addition, this review focus on the list and statistics of previous reports of  seaweeds, not enough critical views on the trends which should give more discussion. This review should be reconsidered after major revision.

Author Response

Thank you for your consideration and we hope we have committed to all your comments about the manuscript.

The manuscript title is Current trends on seaweeds: looking at chemical composition, phytopharmacology and cosmetic applications. The review from this point of view is rather complete.

However, the author spent more time and word on chemical composition and phytopharmacology. It is top-heavy.

Some paragraphs were summarized and modified:

The paragraph (2.3. Effect of global climate changes) was deleted from the section and a summarized idea about the effect of climate changes was added to the introduction section. The paragraph “2.4. Heavy Metals Accumulation” was summarized and added to the paragraph “5.1.A brief overview to regulatory practices”. The paragraph dealing with the use of seaweeds in cosmetic was put at the beginning of the section “6. Targeting seaweeds potentialities for cosmetic purposes” The paragraph “7.1. Consumers’ expectations and current desires” was merged with the paragraph “7. Seaweeds as promising cosmetics: what will happen?” In addition, this review focuses on the list and statistics of previous reports of seaweeds, not enough critical views on the trends which should give more discussion.

Some critical views were cited on the paragraph: 7. Seaweeds as promising cosmetics: what will happen?

Reply: All edits were made in the revised manuscript file.

Reviewer 2 Report

The paper provides a good review of the state of knowledge on potential cosmetic and pharmacological application of seaweeds and it would fit in well in the journal Molecules.

The article deserves publication, but it is necessary that the authors make some revisions, especially for the language (there are many grammar and spelling mistakes). 

Some aspects that should be considered, both in terms of organization of text and structure, are detailed in the attached document

Author Response

Thank you for your consideration and we hope we have committed to all your comments about the manuscript.

The paper provides a good review of the state of knowledge on potential cosmetic and pharmacological application of seaweeds and it would fit in well in the journal Molecules. The article deserves publication, but it is necessary that the authors make some revisions, especially for the language (there are many grammar and spelling mistakes). 

English editing was performed.

Some aspects that should be considered, both in terms of organization of text and structure, are detailed in the attached document.

All the suggested modifications were done and highlighted.

Commented [RV1]: Content and abundance is the same

It was modified “have shown that the content in some secondary metabolites”

Commented [RV3]: Maybe it is better to say “environmental streeses”

Modified as requested

Commented [RV4]: Such as?

“such as the cataloguing of marine chemicals, X-ray and NMR activity.”

Commented [RV5]: Do you mean undiscovered? Yes, unexplored.

Commented [RV6]: Please correct this sentence

“However, there are still numerous unexplored natural bioactive compounds which could serve as a source of novel bioactive compounds for pharmaceutical companies.”

Commented [RV7]: I think that these paragraphs must be moved at the end of the paper together with their medicinal and cosmetic application.

At the end of the review sections focus on the medicinal application of seaweeds; however, these two paragraphs are about the nutritional and fertilizer use of seaweeds which could be as an introduction for the primary and secondary metabolites of seaweeds.

Commented [RV8]: I don’t get the sense of these two paragraphs. Even if you want to give a general critical perspective on seaweeds, these are very vague and not in line with the rest of the review. if you want to put these topics in the paper you need to develop them better. I would suggest removing them and putting something related to these topics just in the conclusion.

The paragraph “2.3. Effect of global climate changes” was summarized and added to the introduction section.

The paragraph “2.4. Heavy Metals Accumulation” was summarized and added to the paragraph “5.1.A brief overview to regulatory practices”.

Commented [RV9]: Please rephrase this sentence. It is not clear

“These polysaccharides have a high commercial value and important applications in the pharmaceutical and food industries [49-51]. They most well-known polysaccharides with industrial applications are the sulphated (fucoindans, carragenans and ulvan) and non-sulphated (e.g. alginates and agars) polysaccharides. Some are less abundant, currently without industrial applications and still under investigation such as laminarin, xylans, porphyrans, argassan and floridean [52].”

Commented [RV10]: Please rephrase and put a citation

“seaweeds structure is more flexible which helps them stand strong against ocean waves and currents [52].”

Commented [RV11]: Please rephrase

“However, the path is to be done, and nowadays the carrageenans are already included in the Britannica, European and American pharmacopoeias and therefore, they will be soon used as pharmaceutical excipients.”

Commented [RV12]: How?

“environment depending on several factors such as salinity, temperature, pH and light.”

Commented [RV13]: Please rephrase

“that must come from the food. “

Commented [RV14]: Please rephrase this part. It is very confusing

“It should be noted that the comparison of different protein contents from various studies is not always easy or feasible, considering the differences of the quantitative extraction procedures [111,113], or different ways of calculations (sum of each amino acid residues content or use of a nitrogen-to-protein conversion factor). Above all, it is the use of different values for nitrogen-to-protein conversion (6.25, mainly used for different types of food; 5, the one recommended to seaweeds by Angell et al. [114]; or 4.59, 5.13 and 5.38 depending on the group to which seaweed belongs [115]).”

Commented [RV15]: This is not clear

“However, it should be noted that a high percentage of essential amino acids doesn’t mean a high concentration of these amino acids in the whole alga, as showed by Angell et al. [114], when compared to foods of high protein content.”

Commented [RV16]: This is quite obvious

It was corrected “The seaweeds proteins are also important as a source of peptides and amino acids extracts, mainly after enzymatic digests”

Commented [RV17]: What does it mean?

It was modified “For example, the peptide obtained from seaweeds protein wastes (Figure 1) exhibits a high antioxidant activity by different mechanisms [120],”

Commented [RV18]: This is obvious

It was modified “Lipids are a group of compounds”

Commented [RV19]: Which one? 10-15%?

0.3-5 % as indicated in table 1

Commented [RV20]: What is low?

The low quantity of lipids and so the beginning of the paragraph was modified as follow: “Beside its low quantity, the chemical characterization of seaweed lipids is the subject of much research, as shown below”.

Commented [RV21]: Please rephrase

“The most abundant lipid class may be phospholipids or glycoglycerolipid [46,122].”

Commented [RV23]: Why?

The surrounding environments appear to have an impact on the genetic expression of the secondary metabolites.

Some secondary metabolites are produced in response to specific environmental conditions.

Innovative methods to the study of secondary metabolites allow us to determine the ability of alga to generate defense compounds in situ, and transport these compounds to specific sites in response to environmental stimuli

Commented [RV24]: I would put this paragraph at the beginning of the discussion about the potential health properties of seaweeds and their medical applications (before the paragraph 4.1)

The paragraph dealing with the use of seaweeds in cosmetic was put at the beginning of the section “6. Targeting seaweeds potentialities for cosmetic purposes”

Commented [RV25]: You already stated these things

They were removed.

Commented [RV26]: I found this paragraph really wordy and not necessary. It could be easily shortened in few sentences in the conclusion section

The paragraph “7.1. Consumers’ expectations and current desires” was merged with the paragraph “7. Seaweeds as promising cosmetics: what will happen?”

Commented [RV27]: Which field?

The exploitation of natural products.

Commented [RV28]: This sentence is incomplete

The sentence was modified: “In an old book [374], a chapter was dedicated to exploit natural product diversity including fragrances and cosmetics of marine originating products. This field has been since then increasingly developed;”

Commented [RV29]: In which sense?

The paragraph was summarized and was put at the beginning of the section “6. Targeting seaweeds potentialities for cosmetic purposes”

Reply: all edits was made in the manuscript file.

Reviewer 3 Report

Interesting work quite well documented although it should be reviewed in order to homogenize the wording of the different paragraphs. The most valuable part of the manuscript is the chapter about secondary metabolites from seaweeds. But in Rhodophyta I should like to have seen something about species of Plocamium or species of the order Bonnemaisoniales like Asparagopsis, Bonnemaisonia, etc. Look for information on this topic! See another changes/corrections below:

Line 43, replace: “…multicellular or benthic marine algae. Marine seaweeds…” by: “multicellular benthic marine algae. Seaweeds…”

Line 46, delete the sentence: “Seaweeds or marine…called as ocean kelps [3]”

Line 47, replace: “They belong to the division Thallophyta in plant kigdom” by: “Seaweeds belong to different kingdoms within the classification of living organisms”

Line 56, replace: “Marine algae are taxonomically classified into… based on the type of pigments” by: “Marine algae are classically divided into four groups, the prokaryotic Cianophyceae (blue green algae) and the eukaryotic Ochrophyta (brown algae), Rhodophyta (red algae) and Chlorophyta (green algae) which differ mainly in their pigmentary composition, nature of their cell walls and reserve polysaccharides”

Line 58, delete the sentence: “Out of these, brown…others contribute 72.8% [5]”

Line 59, replace: “Being seaweeds the primary producers” by: “Beind seaweeds one of the primary producers”

Line 86, replace: “(Laminaria spp.)” by: “(Saccharina japonica)”

Line 87, replace: “(Porphyra spp.)” by: “(Pyropia spp.)”

Line 91, replace: “…a of the genus Eucheuma…” by: “…species of Gelidium…

Paragraph lines 89-103 about polysaccharides: Nothing is said about carrageenans!

Line 107, delete “, especially sodium and iodine.” This clarification makes no sense, mineral compositions is extremely variable and depends of the group and species of seaweeds, e.g. iodine only is abundant in kelps.

Paragraph lines 129-139 (2.3. Effect of global climate changes). This matter has no sense in this kind of review and the selected references are not very relevant for this topic. I think this this section must be deleted.

Paragraph lines 152-154. This paragraph is repetitive (see lines 56-58) and must be deleted.

Lines 169-170 (Table 1). I think is very confused to give the average content of seaweeds primary metabolites by groups (green, brown and red algae) instead of by species, as in the papers of reference. In any case, it would have been better to give the data in the form of intervals. In the current state, the table is cumbersome and not very useful.

Lines 200-201 (Tabe 2). At least the function of agars, carrageenans, ulvans and alginates is not only structural. They also have importance in reproduction processes, as reserve substance in starving periods, to substrate fixation, etc. It would be better to say "mainly structural". Is true that the content and type of carrageenans is species specific and depends of ecophysiological factors, but also of the life cycle phase of the algae in question.

Line 226, Gigartina is a carraagenophyte not an agarophyte.

Line 231-232. Is not true that Gracilaria gracilis is the red algae with the highest agar content and even less with the best quality. Species of Gelidium are much better as far as I know.

Line 252, is indicated thar carrageenans as additive in food industries are know as E 407 and nothing is said about agar (E 406), alginates (E 401 – E 405).

Line 327, replace: “hyperborean” by: “hyperborea”; the same in line 352.

Line 331, replace: “Laminaria japonica” by: “Saccharina japonica”.

Line 362, delete “soil and”; seaweeds are thallophytes without true roots, do not absorb minerals from the substrate.

Line 398, “Mg”?

Line 399, Gracilaria vermiculophylla is now known as Agarophytum vermiculophyllum.

Line 404, hyperborean.

Lines 499 and 500, check citation of references “Miyasita et al., 2013” and “Kendel et al., 2015”

Lines 500 and 501, lipid content depends also of the reproductive state of seaweeds, when Ulva get into reproduction processes forms a lot of mobile spores or gametes rich in lipids. This could be the reason of the high values in Tunisian Ulva lactuca…

Line 792, “sp.” without italics.

Line 793, replace: “pulchara” by: “pulchra”.

Line 797, replace: “pneumonia” by: “pneumoniae”.

Line 807, “sp.” without italics.

Lines 815, 817, Enteromopha linza is now known as Ulva linza.

Line 816, replace: “P. gingivalis” by:”Porphyromonas gingivalis

Line 819, replace: “L. japonica” by: “Saccharina japonica”.

Line 820, replace: “Porphyromonas” by: “P.”

Line 830, delete “Irish”; H. elongata is a common species in many European countries with Atlantic coasts, from Norway to Spain.

Line 836, replace: “S. typhimurium” and “U. lactuca” by: “Salmonella typhimurium” and “Ulva lactuca”, respectively.

Line 837, replace: “E. coli”, “A. nodosum” and “M. luteus” by: “Escherichia coli”, “Ascophyllum nodosum” and “Microccocus luteus”, respectively.

Line 839, “and the brown seaweed…”

Line 845, “sp.” without italics.

Line 853 y 879, replace: “Laminaria” by: “Sacccharina”.

Line 867, “Haligra sp.”; this genus do not exists within the seaweeds! An insect?

Line 883, delete “methanolic”.

Line 888, Lobophora variegata, not variegate.

Line 901, replace: “schroederiare” by “schroederi”.

Line 903, Caulerpa racemosa, not racemose.

Line 904, et al. not in italics.

Line 904, Codium fragile and Sargassum horneri.

Line 920, Undaria pinnatifida, Porphyridium sp., Phaeodactylum sp.

Line 922, “fucoidan”, not “Fucoidan”.

Line 951, replace “E. kurome Okamura” by: “Ecklonia kurome”. The same in table 3.

Line 952, “Ascophyllum nodosum” and “Saccharina japonica”. The same in table 3.

Line 953, replace “S. coreanum J. Agardh” by: “Sargassum coreanum”. The same in table 3.

Line 954, replace “E. cava Kjellman” by: “Ecklonia cava”. The same in table 3.

Line 956, “Eisenia bicyclis” and “Undaria pinnatifida”. The same in table 3.

Line 970, replace:“Laminaria japonica” by “Saccharina japonica”.

Line 973, “Spatoglossum schroederi”. The same in table 3.

Line 973, replace: “from Arthrospira pratensis” by: “from the blue-green algae Arthospira pratensis”. The same in table 3.

Lines 1005-1006, “elevated levels… may constitute a health risk… deleterious health effects”. Very alarming phrase based on a little rigorous reference (318).

Line 1012, replace: “Laminaria saccharina” by: “ Saccharina latissimi”.

Line 1013, Fucus vesiculosus”; delete: Enteromorpha spp. (all species of Enteromorpha are now Ulva species).

Line 1014, replace: “Porphyra tenera, Porphyra yezoensis, Porphyra dioica, Porphyra purpurea, Porphyra laciniata, Porphyra leucosticta, Gracilaria verrucosa and Lithothamnium calcareum” by: “Pyropia tenera, P. yezoensis, P. leucosticta, Porphyra dioica, P. purpurea, P. laciniata, Gracilaria gracilis and Phymatolithon calcareum”.

Line 1019, “Clostridium perfingrens”, in italics.

Lines 1070, 1073, 1156 and 1161, replace: “Sargassum” by: “Sargassum sp.”.

Line 1177, replace: “Laminaria saccharina” by: “Saccharina latissima”.

Lines 1295-1296, Wouldn't it be better to put the chemical symbols?

Author Response

Thank you for your consideration and we hope we have committed to all your comments about the manuscript.

Comments:

Interesting work quite well documented although it should be reviewed in order to homogenize the wording of the different paragraphs.

The manuscript was reviewed and modifications were performed.

The most valuable part of the manuscript is the chapter about secondary metabolites from seaweeds. But in Rhodophyta I should like to have seen something about species of Plocamiumor species of the order Bonnemaisoniales like AsparagopsisBonnemaisonia, etc. Look for information on this topic!

Two paragraphs were added as per your request.

“Different species of the family Bonnemaisoniaceae are well known as sources of halogenated compounds [193] with strong antifungal and antibiotic activity [193].”

Plocamium cartilagineum is a species of red algae (family Plocamiaceae, order Gigartinales). This species is characterized by its interesting secondary metabolites, being a rich source of diverse polyhalogenated monoterpenes, with a surprising degree of halogen incorporation [207].”

See another changes/corrections below:

Line 43, replace: “…multicellular or benthic marine algae. Marine seaweeds…” by: “multicellular benthic marine algae. Seaweeds…” Line 46, delete the sentence: “Seaweeds or marine…called as ocean kelps [3]” Line 47, replace: “They belong to the division Thallophyta in plant kigdom” by: “Seaweeds belong to different kingdoms within the classification of living organisms” Line 56, replace: “Marine algae are taxonomically classified into… based on the type of pigments” by: “Marine algae are classically divided into four groups, the prokaryotic Cianophyceae (blue green algae) and the eukaryotic Ochrophyta (brown algae), Rhodophyta (red algae) and Chlorophyta (green algae) which differ mainly in their pigmentary composition, nature of their cell walls and reserve polysaccharides” Line 58, delete the sentence: “Out of these, brown…others contribute 72.8% [5]” Line 59, replace: “Being seaweeds the primary producers” by: “Beind seaweeds one of the primary producers” Line 86, replace: “(Laminaria)” by: “(Saccharina japonica)” Line 87, replace: “(Porphyra)” by: “(Pyropiaspp.)” Line 91, replace: “…a of the genus Eucheuma…” by: “…species of Gelidium…

All the above suggested modifications (3-11) were performed and highlighted as per your request.

Paragraph lines 89-103 about polysaccharides: Nothing is said about carrageenans!

A paragraph was added between lines 94 and 97.

Carrageenans are a family of linear sulfated polysaccharides widely used in food industry and are extracted from red edible seaweeds. Seaweed carrageenan and agar are used in food industry as thickening and gelling agents, they are also used as growth medium for microorganisms [21].

Line 107, delete “, especially sodium and iodine.” This clarification makes no sense, mineral compositions is extremely variable and depends of the group and species of seaweeds, e.g. iodine only is abundant in kelps.

It was modified as per your request.

Paragraph lines 129-139 (2.3. Effect of global climate changes). This matter has no sense in this kind of review and the selected references are not very relevant for this topic. I think this this section must be deleted.

The paragraph was deleted from the section and a summarized idea about the effect of climate changes was added to the introduction section as follow: “Alterations in global climate and its variability influences the biological, ecological and socioeconomic systems therefore it has remarkable impact on the marine environment as well [10]. Although seaweeds are usually well adapted to their thermal environment, but temperature influences the enzymatic and metabolic functioning of seaweeds [11] and could result in cellular and sub cellular damage [12] leading to slow growth and development [13]. “

Paragraph lines 152-154. This paragraph is repetitive (see lines 56-58) and must be deleted.

It was deleted as per your request.

Lines 169-170 (Table 1). I think is very confused to give the average content of seaweeds primary metabolites by groups (green, brown and red algae) instead of by species, as in the papers of reference. In any case, it would have been better to give the data in the form of intervals. In the current state, the table is cumbersome and not very useful.

Table 1 was modified and data was given in the form of intervals.

Lines 200-201 (Tabe 2). At least the function of agars, carrageenans, ulvans and alginates is not only structural. They also have importance in reproduction processes, as reserve substance in starving periods, to substrate fixation, etc. It would be better to say "mainly structural". Is true that the content and type of carrageenans is species specific and depends of ecophysiological factors, but also of the life cycle phase of the algae in question.

It was modified and the expression “mainly structural” was used.

Line 226, Gigartinais a carraagenophyte not an agarophyte.

It was modified.

Line 231-232. Is not true that Gracilaria gracilis is the red algae with the highest agar content and even less with the best quality. Species of Gelidium are much better as far as I know.

A reference was added about Gelidium as follow: “Other researchers suggest that Gelidium-extracted agar typically has better quality, such as higher gel strength [68].”

Line 252, is indicated thar carrageenans as additive in food industries are know as E 407 and nothing is said about agar (E 406), alginates (E 401 – E 405).

It was added on lines 211 and 235.

Line 327, replace: “hyperborean” by: “hyperborea”; the same in line 352. Line 331, replace: “Laminaria japonica” by: “Saccharina japonica”. Line 362, delete “soil and”; seaweeds are thallophytes without true roots, do not absorb minerals from the substrate.

All the above suggested modifications (21-23) were performed and highlighted as per your request.

Line 398, “Mg”?

This is about Mn content as per references.

The content of Mg was indicated in table 1.

Line 399, Gracilaria vermiculophyllais now known as Agarophytum vermiculophyllum.

It was modified.

Line 404, 

It was modified.

Lines 499 and 500, check citation of references “Miyasita et al., 2013” and “Kendel et al., 2015”

They were checked: “Miyasita et al., 2013” is reference 122, “Kendel et al., 2015” is reference 44.

Lines 500 and 501, lipid content depends also of the reproductive state of seaweeds, when Ulvaget into reproduction processes forms a lot of mobile spores or gametes rich in lipids. This could be the reason of the high values in Tunisian Ulva lactuca…

It was modified.

Line 792, “” without italics. Line 793, replace: “pulchara” by: “pulchra”. Line 797, replace: “pneumonia” by: “pneumoniae”. Line 807, “” without italics. Lines 815, 817, Enteromopha linzais now known as Ulva linza. Line 816, replace: “ gingivalis” by:”Porphyromonas gingivalis” Line 819, replace: “ japonica” by: “Saccharina japonica”. Line 820, replace: “Porphyromonas” by: “ Line 830, delete “Irish”;  elongatais a common species in many European countries with Atlantic coasts, from Norway to Spain. Line 836, replace: “ typhimurium” and “U. lactuca” by: “Salmonella typhimurium” and “Ulva lactuca”, respectively. Line 837, replace: “ coli”, “A. nodosum” and “M. luteus” by: “Escherichia coli”, “Ascophyllum nodosum” and “Microccocus luteus”, respectively. Line 839, “and the brown seaweed…” Line 845, “” without italics. Line 853 y 879, replace: “Laminaria” by: “Sacccharina”.

All the above suggested modifications (29-42) were performed and highlighted as per your request.

Line 867, “Haligra”; this genus do not exists within the seaweeds! An insect?

It was deleted.

Line 883, delete “methanolic”. Line 888, Lobophora variegata, not  Line 901, replace: “schroederiare” by “schroederi”. Line 903, Caulerpa racemosa, not  Line 904, et al.not in italics. Line 904, Codium fragileand Sargassum horneri. Line 920, Undaria pinnatifidaPorphyridiumPhaeodactylumsp. Line 922, “fucoidan”, not “Fucoidan”. Line 951, replace “ kurome Okamura” by: “Ecklonia kurome”. The same in table 3. Line 952, “Ascophyllum nodosum”and “Saccharina japonica”. The same in table 3. Line 953, replace “ coreanumJ. Agardh” by: “Sargassum coreanum”. The same in table 3. Line 954, replace “ cava Kjellman” by: “Ecklonia cava”. The same in table 3. Line 956, “Eisenia bicyclis” and “Undaria pinnatifida”. The same in table 3. Line 970, replace:“Laminaria japonica”by “Saccharina japonica”. Line 973, “Spatoglossum schroederi”. The same in table 3. Line 973, replace: “from Arthrospira pratensis” by: “from the blue-green algae Arthospira pratensis”. The same in table 3.

All the above suggested modifications (44-59) were performed and highlighted as per your request.

Lines 1005-1006, “elevated levels… may constitute a health risk… deleterious health effects”. Very alarming phrase based on a little rigorous reference (318).

The sentence was modified.

Line 1012, replace: “Laminaria saccharina” by: “ Saccharina latissimi”. Line 1013, Fucus vesiculosus”; delete: Enteromorpha (all species of Enteromorphaare now Ulva species). Line 1014, replace: “Porphyra tenera, Porphyra yezoensis, Porphyra dioica, Porphyra purpurea, Porphyra laciniata, Porphyra leucosticta, Gracilaria verrucosaand Lithothamnium calcareum” by: “Pyropia tenera, P. yezoensis, P. leucosticta, Porphyra dioica, P. purpurea, P. laciniata, Gracilaria gracilis and Phymatolithon calcareum”. Line 1019, “Clostridium perfingrens”, in italics. Lines 1070, 1073, 1156 and 1161, replace: “Sargassum” by: “Sargassum”. Line 1177, replace: “Laminaria saccharina” by: “Saccharina latissima”.

All the above suggested modifications (61-66) were performed and highlighted as per your request.

Lines 1295-1296, Wouldn't it be better to put the chemical symbols?

They were deleted as they were mentioned before in the section “3.1.2. Minerals”

Reply: all edits was made in the manuscript file.

Round 2

Reviewer 1 Report

This is much improved manuscript. I fully support its publication.